# A randomised controlled trial to reduce highest priority critically important antimicrobial prescription in companion animals

David A. Singleton [1✉], Angela Rayner[2], Bethaney Brant[1], Steven Smyth[1], Peter-John M. Noble[1], Alan D. Radford[1] & Gina L. Pinchbeck[1]

Robust evidence supporting strategies for companion animal antimicrobial stewardship is limited, despite frequent prescription of highest priority critically important antimicrobials (HPCIA). Here we describe a randomised controlled trial where electronic prescription data were utilised (August 2018–January 2019) to identify above average HPCIA-prescribing practices ($n = 60$), which were randomly assigned into a control group (CG) and two intervention groups. In March 2019, the light intervention group (LIG) and heavy intervention group (HIG) were notified of their above average status, and were provided with educational material (LIG, HIG), in-depth benchmarking (HIG), and follow-up meetings (HIG). Following notification, follow-up monitoring lasted for eight months (April–November 2019; post-intervention period) for all intervention groups, though HIG practices were able to access further support (i.e., follow-up meetings) for the first six of these months if requested. Post-intervention, in the HIG a 23.5% and 39.0% reduction in canine (0.5% of total consultations, 95% confidence interval, 0.4-0.6, $P = 0.04$) and feline (4.4%, 3.4-5.3, $P < 0.001$) HPCIA-prescribing consultations was observed, compared to the CG (dogs: 0.6%, 0.5-0.8; cats: 7.4%, 6.0-8.7). The LIG was associated with a 16.7% reduction in feline HPCIA prescription (6.1% of total consultations, 5.3-7.0, $P = 0.03$). Therefore, in this trial we have demonstrated effective strategies for reducing veterinary HPCIA prescription.

[1] Institute of Infection, Veterinary and Ecological Sciences, University of Liverpool, Leahurst Campus, Chester High Road, Neston, UK. [2] CVS (UK) Limited, 1 Owen Road, Diss, UK. ✉email: D.A.Singleton@liverpool.ac.uk

Companion animals are increasingly being recognised as an important contributor to the development[1,2], carriage[3] and transmission of antimicrobial resistant (AMR) bacteria both between animals and to/from humans, primarily due to frequent antimicrobial prescription driving selection for resistance in both species groups[1,4–8], and the close proximity in which companion animals reside with humans[8,9]. Recent population studies facilitated by the expanding availability of companion animal electronic health records (EHRs)[10,11] have identified encouraging trends in antimicrobial prescription, including reducing antimicrobial frequency, both in general[10] and for some clinical presentations[12–14]. However, antimicrobials still remain amongst the most commonly prescribed pharmaceutical agents in companion animals[15].

Of particular concern, cefovecin, a 3rd generation cephalosporin considered a "highest priority critically important antimicrobial" (HPCIA) by the World Health Organisation[16], remains the most commonly prescribed antimicrobial in cats[10,11]; such prescriptions frequently lacking clearly recorded clinical reasoning to justify their prescription[17]. HPCIAs, also including fluoroquinolones which are also relatively frequently prescribed to companion animals[10], are recommended to ideally be reserved for human use alone[16], with current companion animal prescribing guidance suggesting HPCIAs should only be prescribed when there is clear evidence of resistance to first-line antimicrobials[18]. As cefovecin is frequently prescribed in the absence of such evidence[17], this raises significant questions as to how appropriate such frequent prescription is in cats.

Along with more qualitative studies[19–23], EHRs[17,24,25] have also identified key motivators for antimicrobial prescription, revealing a complex interplay between animal, owner, clinical presentation, individual veterinary surgeon, and the overarching culture of the veterinary practice in which they are employed. On a population-level, considerable inter-practice variation in antimicrobial prescription frequency, including the HPCIAs, has been identified[10], with those practices with higher levels of professional accreditation being relatively less frequent prescribers of systemically-administered antimicrobials to dogs, compared to their non-accredited peers[25].

Much of the work to mitigate the companion animal contribution to AMR has focused on improving antimicrobial prescription, under the banner of antimicrobial stewardship, largely through the production of evidence-based antimicrobial prescribing guidance[18,26–29] and practice benchmarking[30,31]. Although these are to be welcomed, robust evidence supporting their impact on companion animal antimicrobial prescribing practices is absent. In this regard, there is much to learn from medical practice, where robustly evidenced antimicrobial stewardship schemes have been established for some time[32–38]. In the absence of evidence, individuals tend to over-estimate perceived negative traits in their peers, thus serving as justification for their own behaviour (e.g., alcohol consumption frequency)[39]. A similar tendency regarding antimicrobial prescription amongst veterinary surgeons has been previously observed[19], leading us to hypothesise that lack of knowledge of relative frequencies of antimicrobial prescription might in itself serve as a driver for more frequent prescription. As such, use of benchmarking, thus drawing attention to an individual's departure from a "social norm", might have some utility in counteracting such tendencies. Indeed, use of prescribing benchmarks within a social norms framework has already shown some potential in prompting medical general practitioner-reflection and behavioural change[34].

Hence, in this work we present a randomised controlled trial demonstrating the efficacy of social norm messaging, via integrated EHR-driven antimicrobial prescription benchmarking and in-practice educational support, to significantly reduce HPCIA prescription frequency, and antimicrobial prescription frequency in general. Participants were recruited from a cohort of above average HPCIA-prescribing practices within a single veterinary practice group (CVS Group Ltd.), with the hypothesis that such interventions would effectively reduce HPCIA prescription frequency post-intervention.

## Results

**Heavy intervention group trial engagement.** Following above average total HPCIA prescription frequency notification (28-29 March 2019), HIG practices were given the option to voluntarily participate in a further reflection and education programme; two HIG practices opted out (Fig. 1). As all practices had previously consented to routinely provide data, data continued to be collected from all practices both before and following intervention, irrespective of their active participation in this trial. All 18 consenting HIG practices participated in an initial review with a hub clinical lead (15 held in April, two in May and one in June 2019), and 16 held a separate practice meeting (held April–June 2019). Of these, 15 requested a hub clinical lead follow-up review (held April–July 2019), seven requested a further hub clinical lead follow-up review (held May–July 2019), and 16 held a final hub clinical lead review (held August–October 2019). No significant variations in baseline characteristics between groups were observed (Table 1).

**Antimicrobial prescription frequency.** Pre-intervention, 0.71% (95% confidence interval, CI, 0.57–0.86, $n$ consultations = 88,298), 0.72% (CI 0.60–0.83, $n$ consultations = 107,223), and 0.68% (CI 0.47–0.90, $n$ consultations = 63,366) of canine consultations were associated with total HPCIA prescription in the CG, LIG and HIG, respectively (Table 2). No significant variations between intervention groups were observed ($P = 0.95$). In cats, 7.44% CG (CI 6.27-8.61, $n$ consultations = 33,613), 7.03% LIG (CI 6.32–7.74, $n$ consultations = 39,803), and 8.03% HIG (CI 7.16–8.90, $n$ consultations = 25,661) consultations were associated with total HPCIA prescription (Table 3). No significant variations between intervention groups were observed ($P = 0.48$). The "respiratory", "kidney disease" and "pruritus" MPCs in dogs, and "respiratory" and "trauma" MPCs in cats were associated with increased total HPCIA prescription frequency, relative to other MPCs.

Post-intervention, HIG canine total HPCIA prescription frequency significantly reduced by 23.5% (0.49% of canine HIG consultations, CI 0.37–0.61, $P = 0.04$, $n$ consultations = 63,301), but no significant change was observed in the LIG (0.79% of canine LIG consultations, CI 0.61-0.97, $P = 0.78$, $n$ consultations = 105,518), compared to the CG (0.64% of canine CG consultations, CI 0.47–0.82, $n$ consultations = 81,582) (Table 2; full model results Supplementary Table 2). For cats, a significant 39.0% and 16.7% decrease in HPCIA prescription frequency was observed in both the HIG (4.35% of feline HIG consultations, CI 3.41–5.29, $P < 0.01$, $n$ consultations = 25,808), and LIG (6.14% of feline LIG consultations, CI 5.29-7.00, $P = 0.03$, $n$ consultations = 37,339), respectively, compared to the CG (7.37% of feline CG consultations, CI 6.02–8.72, $n$ consultations = 30,362) (Table 3; full model results Supplementary Table 3). Both models provided a good fit to underlying data, with the fixed terms explaining 10% of canine and 40% of feline variance (Supplementary Table 2 and 3). Fourteen and twenty HIG practices recorded post-intervention total HPCIA prescription frequency decreases in dogs (Fig. 2A) and cats (Fig. 3A), respectively. In the LIG, nine and sixteen practices were associated with post-intervention HPCIA prescription frequency reductions in dogs and cats respectively, whereas in the CG, ten

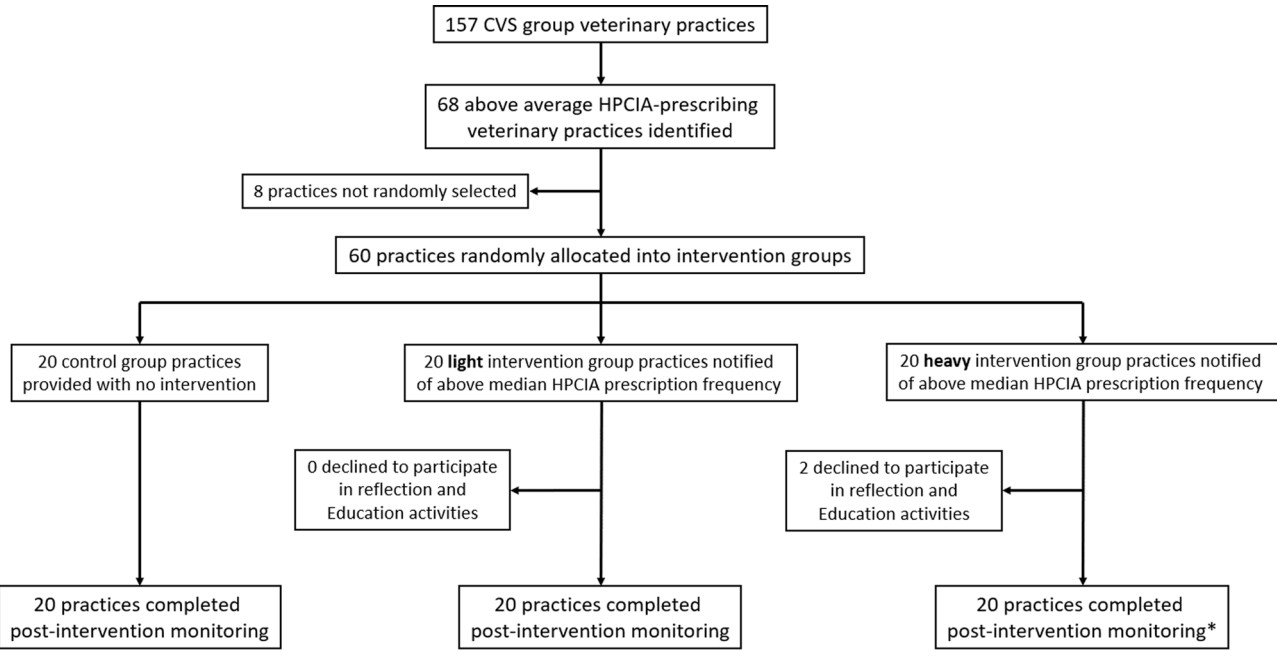

**Fig. 1 Trial practice group allocation.** Schematic diagram of the process used to select above average highest priority critically important antimicrobial (HPCIA) prescribing veterinary practices owned by CVS Group Limited, followed by drop-off rate following the initial benchmarking intervention.

practices were associated with post-intervention HPCIA prescription frequency reductions in both species.

Regarding HIG month-by-month variation, significant reductions in total HPCIA prescription frequency were observed for three and eight post-intervention months in dogs (Table 4; Fig. 2B; full model results Supplementary Table 4) and cats (Table 4; Fig. 3B; full model results Supplementary Table 5), respectively, with a particularly steep decline being observed between March and April 2019 in cats. The LIG was associated with a significant HPCIA prescription frequency reduction in cats in June and November 2019, compared to the CG. Addition of temporal fixed terms was found to improve model fit (Supplementary Table 4 and 5). Overall, no significant temporal variation in HPCIA prescription frequency was observed in the CG in either species, whereas a significant linear reduction was observed in both dogs ($P = 0.005$) and cats ($P < 0.001$) in the HIG. Whereas in the LIG a significant HPCIA prescription frequency linear reduction was observed for cats ($P = 0.01$), no significant trend was observed for dogs (full model results Supplementary Table 6).

In HIG dogs, HPCIA prescription frequency was found to significantly reduce in one MPC post-intervention (Table 2; full model results Supplementary Table 7), whereas three MPCS were associated with reductions in cats (Table 3; full model results Supplementary Table 8). In the LIG for dogs, no MPCS showed significant reductions in HPCIA prescription frequency, although the "vaccination" MPC was associated with a significant increase. In cats in the LIG no MPCs showed a significant reduction in HPCIA prescription frequency. Only the "other healthy" and "gastroenteric" MPCs in dogs were associated with improved fit compared to a null model in dogs, whereas six MPCs were associated with improved fits in cats.

Considering antimicrobial prescription more broadly, the HIG was also associated with a 18.9% and 17.3% significant decrease in the percentage of dog (Table 2; full model results Supplementary Table 2) and cat (Table 3; full model results Supplementary Table 3) consultations prescribed a systemic antimicrobial, compared to the

CG, respectively. Significant reductions in total antimicrobial prescription frequency were also observed in both species in the HIG. In terms of other pharmaceutical agents, we found no significant post-intervention group differences in anti-inflammatory prescription frequency, nor frequency of euthanasia in either species (dogs, Table 2, Supplementary Table 9; cats, Table 3, Supplementary Table 9).

**Antimicrobial prescription choice.** A summary of antimicrobial prescription choice variation as a percentage of total antimicrobial prescriptions by class, and for beta-lactams by sub-class, is available in Table 5 (dogs; full model results Supplementary Table 10) and Table 6 (cats; full model results Supplementary Table 11), and as a percentage of total consultations in Supplementary Table 12 (dogs) and 13 (cats). Though no comparisons were significant, there was some evidence of post-intervention increases in clavulanic acid potentiated amoxicillin prescription in cats in the HIG, overtaking 3rd generation cephalosporins as the most prescribed antimicrobial.

**Bacterial diagnostic test frequency.** Pre-intervention there were 3,932 cytological test orders, representing 1.09% of CG (CI 0.67–1.51), 1.03% LIG (CI 0.47–1.58) and 1.18% HIG (CI 0.54–1.83) consultations. Of 4,783 post-intervention cytological test orders, 1.29% of CG (CI 0.77–1.81), 1.31% LIG (CI 0.51–2.11) and 1.60% HIG (CI 1.03–2.17) post-intervention consultations were associated with an order. However, no significant increase was observed, compared to the CG (LIG $P = 0.53$; HIG $P = 0.89$).

Pre-intervention, there were 4517 bacterial culture and susceptibility test orders, representing 1.16% of CG (CI 0.77–1.55), 1.01% LIG (CI 0.64–1.39) and 1.76% HIG (CI 1.05–2.47) consultations. Of 4623 post-intervention bacterial culture and susceptibility test orders, 1.19% of CG (CI 0.83–1.55), 1.12% LIG (CI 0.68–1.55) and 1.81% HIG (CI 1.20–2.42) post-intervention consultations were associated with an order. However, no significant increase was observed, compared to the CG (LIG $P = 0.94$; HIG $P = 0.20$). Full

**Table 1 Practice, canine and feline pre-intervention (August 2018–March 2019 inclusive) baseline characteristics.**

| Variable | Control group | Light intervention group | Heavy intervention group | P |
|---|---|---|---|---|
| Practice characteristics | | | | |
| Median vet FTE[a]/practice [range] | 3.8 [1.1–17.4] | 4.1 [1.0–13.4] | 5.0 [1.1–11.5] | 0.59 |
| % of total FTE locum cover | 6.5 | 5.0 | 6.8 | 0.24 |
| CANINE | | | | |
| Median n consultations/practice [range] | 2,657.5 [460.0–23,889.0] | 3,814.5 [1,004.0–13,476.0] | 3,127.0 [443.0–9,145.0] | 0.17 |
| Median n unique animals/practice [range] | 1,442.5 [335.0–14,436.0] | 2,104.0 [626.0–6,613.0] | 1,745.5 [310.0–4,785.0] | 0.21 |
| Main presenting complaint—% of total consultations (95% confidence interval, CI) | | | | |
| Vaccination | 32.5 (29.5–35.4) | 29.4 (26.9–31.9) | 33.5 (29.9–37.0) | 0.61 |
| Other healthy | 21.8 (17.7–26.0) | 27.7 (23.6–31.7) | 21.7 (18.7–24.7) | 0.66 |
| Post-operative check | 9.8 (6.8–12.8) | 8.4 (7.0–9.8) | 7.1 (6.1–8.2) | 0.46 |
| Gastroenteric | 3.0 (2.4–3.7) | 3.2 (2.7–3.7) | 3.1 (2.5–3.7) | 0.31 |
| Respiratory | 1.0 (0.7–1.2) | 0.9 (0.8–1.0) | 1.0 (0.7–1.2) | 0.29 |
| Pruritus | 4.8 (3.8–5.9) | 4.7 (3.7–5.6) | 4.7 (3.9–5.6) | 0.33 |
| Trauma | 4.2 (3.3–5.0) | 4.4 (3.7–5.0) | 4.4 (3.5–5.3) | 0.31 |
| Tumour | 1.4 (1.1–1.7) | 1.5 (1.2–1.7) | 1.7 (1.3–2.0) | 0.12 |
| Kidney disease | 0.3 (0.2–0.3) | 0.3 (0.2–0.4) | 0.3 (0.2–0.4) | 0.15 |
| Other unwell | 21.3 (18.1–24.5) | 19.6 (16.6–22.7) | 22.4 (20.0–24.7) | 0.94 |
| Animal characteristics—% of total consultations (95% CI) | | | | |
| Sex: Male | 51.2 (50.4–51.9) | 51.0 (49.9–52.1) | 52.5 (51.3–53.8) | 0.17 |
| Neutered | 67.4 (65.4–69.4) | 66.4 (62.2–70.6) | 67.9 (65.1–70.8) | 0.81 |
| Insured | 34.9 (21.9–48.0) | 31.6 (22.2–41.0) | 25.2 (15.7–34.8) | 0.88 |
| Vaccinated | 79.8 (78.1–81.6) | 79.7 (77.6–81.8) | 81.9 (79.4–84.3) | 0.29 |
| Median age [range] | 5.8 [0.0–20.7] | 5.9 [0.0–24.7] | 6.2 [0.0–24.0] | 0.62 |
| FELINE | | | | |
| Median n consultations/practice [range] | 868.0 [231.0–8,608.0] | 1,609.0 [424.0–6,365.0] | 1,062.0 [457.0–2,626.0] | 0.34 |
| Median n unique animals/practice [range] | 577.0 [161.0–5,996.0] | 1,031.0 [340.0–3,922.0] | 740.0 [373.0–1,895.0] | 0.32 |
| Main presenting complaint—% of total consultations (95% CI) | | | | |
| Vaccination | 36.7 (33.0–40.5) | 33.9 (31.7–36.2) | 37.7 (33.9–41.4) | 0.76 |
| Other healthy | 20.4 (16.6–24.2) | 27.8 (23.7–31.9) | 21.7 (18.3–25.1) | 0.62 |
| Post-operative check | 7.9 (5.1–10.7) | 6.6 (5.4–7.9) | 5.8 (5.0–6.6) | 0.50 |
| Gastroenteric | 2.3 (1.6–3.0) | 2.2 (1.8–2.5) | 2.2 (1.7–2.7) | 0.44 |
| Respiratory | 1.4 (1.0–1.9) | 1.2 (0.8–1.6) | 1.4 (1.1–1.8) | 0.36 |
| Pruritus | 2.5 (1.8–3.2) | 2.0 (1.6–2.5) | 2.3 (1.9–2.7) | 0.43 |
| Trauma | 4.6 (3.5–5.7) | 4.6 (3.7–5.5) | 4.2 (3.6–4.8) | 0.33 |
| Tumour | 0.7 (0.5–0.9) | 0.9 (0.6–1.2) | 0.9 (0.7–1.2) | 0.43 |
| Kidney disease | 0.8 (0.5–1.0) | 0.7 (0.6–0.9) | 1.1 (0.7–1.4) | 0.10 |
| Other unwell | 22.6 (19.3–25.9) | 20.1 (17.6–22.6) | 22.7 (20.4–25.1) | 0.94 |
| Animal characteristics—% of total consultations (95% CI) | | | | |
| Sex: Male | 48.5 (47.5–49.5) | 49.0 (47.8–50.1) | 49.6 (48.5–50.7) | 0.38 |
| Neutered | 81.6 (79.0–84.3) | 83.0 (80.9–85.0) | 80.0 (77.1–82.9) | 0.35 |
| Insured | 24.9 (11.1–38.6) | 21.7 (14.2–29.2) | 17.3 (12.1–22.5) | 0.76 |
| Vaccinated | 68.9 (65.6–72.2) | 71.5 (68.1–74.9) | 72.0 (69.2–74.7) | 0.39 |
| Median age [range] | 6.7 [0.0–26.7] | 7.3 [0.0–27.0] | 7.3 [0.0–26.8] | 0.57 |

Practice characteristics summarised as of March 2019; statistical comparisons between groups performed via Kruskal-Wallis tests. Data is derived from 88,298 canine and 33,613 feline pre-intervention CG consultations, 107,223 canine and 39,803 feline pre-intervention LIG consultations, and 63,366 canine and 25,661 pre-intervention HIG consultations.
[a]Full-time equivalent (40 h), inclusive of locum veterinary surgeon FTE.

regression model results for both comparisons are available in Supplementary Table 9; neither model provided improved fit compared to a null model.

**Use of antimicrobial prescription benchmarking portal.** Prior to the intervention (1st February–27th March 2019 inclusive), there was no significant variation between practices logging into their antimicrobial benchmarking portal: CG ($n = 0$), LIG ($n = 3$) and HIG ($n = 3$) (Fisher's Exact Test, $P = 0.23$). Post-intervention (1st April – 30th November 2019 inclusive) however, significant variation was observed between practices logging into the portal within the CG ($n = 3$), LIG ($n = 8$) and HIG ($n = 16$), respectively (Fisher's Exact Test, $P < 0.001$), with significant variation being observed between both the CG and HIG ($P < 0.001$),

and LIG and HIG ($P = 0.03$), but not between the CG and LIG ($P = 0.16$). The two HIG practices that declined further participation did not interact with their portal pre- or post-intervention. A pronounced increase in LIG and HIG practice portal engagement was observed in April 2019, declining towards the end of the trial (Fig. 4).

**Discussion**
Here we describe use of EHR data from veterinary practices to initiate voluntary antimicrobial prescribing behavioural change in veterinary prescribing. In so doing, we have outlined a data-led and educational support framework by which HPCIA prescription frequency can be significantly reduced, while preserving clinical autonomy. These insights are now being used to inform a national

**Table 2 Canine antimicrobial and anti-inflammatory prescription, and euthanasia as a percentage of total canine consultations by intervention group, split into pre-intervention (August 2018–March 2019) and post-intervention (April–November 2019) phases.**

| CANINE Variable | Control group | | Light intervention group | | | Heavy intervention group | | |
|---|---|---|---|---|---|---|---|---|
| | Pre-intervention | Post-intervention | Pre-intervention | Post-intervention | P | Pre-intervention | Post-intervention | P |
| **Primary outcome** | | | | | | | | |
| HPCIA[a] | 0.71 (0.57–0.86) | 0.64 (0.47–0.82) | 0.72 (0.60–0.83) | 0.79 (0.61–0.97) | 0.777 | 0.68 (0.47–0.90) | **0.49 (0.37–0.61)**[b] | **0.044** |
| **Antimicrobial prescription—% of total consultations (95% confidence interval, CI)** | | | | | | | | |
| Total | 17.44 (16.23–18.65) | 17.48 (16.02–18.95) | 17.44 (16.40–18.48) | 17.40 (16.33–18.46) | 0.143 | 17.28 (16.20–18.36) | **15.43 (13.91–16.95)** | **0.0002** |
| Systemic | 10.49 (9.56–11.42) | 10.51 (9.45–11.58) | 10.32 (9.43–11.22) | 10.00 (9.05–10.95) | 0.110 | 10.25 (9.33–11.16) | **8.42 (7.41–9.43)** | **0.0001** |
| Topical | 7.58 (6.95–8.22) | 7.57 (6.83–8.31) | 7.79 (7.39–8.18) | 7.98 (7.61–8.36) | 0.912 | 7.61 (7.11–8.11) | 7.44 (6.61–8.27) | 0.173 |
| Systemic HPCIA | 0.40 (0.29–0.52) | 0.32 (0.21–0.43) | 0.39 (0.34–0.44) | 0.36 (0.31–0.42) | 0.575 | 0.51 (0.35–0.67) | 0.22 (0.15–0.30) | 0.099 |
| **HPCIA prescription by main presenting complaint—% of relevant consultations (95% CI)** | | | | | | | | |
| Vaccination | 0.10 (0.06–0.14) | 0.13 (0.06–0.19) | 0.15 (0.10–0.19) | **0.21 (0.11–0.31)** | **0.012** | 0.12 (0.04–0.20) | 0.04 (0.02–0.06) | 0.088 |
| Other healthy | 0.70 (0.45–0.95) | 0.72 (0.37–1.07) | 0.47 (0.30–0.63) | 0.71 (0.49–0.94) | 0.285 | 0.43 (0.27–0.59) | **0.31 (0.19–0.43)** | **0.008** |
| Post-operative check | 0.75 (0.15–1.36) | 0.65 (0.21–1.10) | 0.62 (0.38–0.87) | 0.66 (0.42–0.89) | 0.305 | 0.53 (0.26–0.81) | 0.33 (0.17–0.49) | 0.869 |
| Gastroenteric | 0.60 (0.25–0.95) | 0.09 (0.00–0.21) | 0.59 (0.30–0.88) | 0.40 (0.14–0.66) | 0.304 | 1.52 (0.37–2.67) | 0.39 (0.00–0.77) | 0.616 |
| Respiratory | 2.05 (0.67–3.43) | 1.56 (0.17–2.96) | 1.88 (0.99–2.78) | 0.73 (0.23–1.22) | 0.491 | 2.34 (0.87–3.80) | 0.34 (0.00–0.97) | 0.295 |
| Pruritus | 1.69 (0.87–2.51) | 1.23 (0.60–1.86) | 2.39 (1.57–3.22) | 2.23 (1.56–2.90) | 0.645 | 1.30 (0.57–2.04) | 1.20 (0.87–1.53) | 0.764 |
| Trauma | 0.55 (0.25–0.86) | 0.25 (0.06–0.45) | 0.49 (0.24–0.74) | 0.62 (0.32–0.93) | 0.232 | 0.61 (0.36–0.87) | 0.60 (0.30–0.90) | 0.260 |
| Tumour | 0.71 (0.38–1.04) | 0.68 (0.19–1.16) | 0.62 (0.11–1.14) | 0.30 (0.00–0.60) | 0.130 | 0.87 (0.25–1.48) | 0.15 (0.00–0.33) | 0.109 |
| Kidney disease | 2.06 (0.16–3.97) | 1.03 (0.00–2.52) | 1.22 (0.00–2.59) | 2.11 (0.56–3.65) | 0.997 | 4.65 (1.46–7.83) | 1.60 (0.37–2.83) | 0.989 |
| Other unwell | 1.39 (1.11–1.68) | 1.32 (1.05–1.58) | 1.58 (1.08–2.08) | 1.60 (1.02–2.17) | 0.981 | 1.44 (0.88–2.00) | 1.20 (0.89–1.50) | 0.354 |
| **Other prescriptions—% of total consultations (95% CI)** | | | | | | | | |
| Anti-inflammatory | 20.26 (18.99–21.54) | 20.62 (19.46–21.79) | 19.91 (18.86–20.95) | 19.89 (18.16–21.02) | 0.977 | 21.41 (19.90–22.92) | 21.05 (19.53–22.57) | 0.668 |
| Euthanasia | 0.99 (0.80–1.18) | 1.04 (0.81–1.26) | 1.09 (0.90–1.29) | 1.16 (0.94–1.38) | 0.541 | 1.07 (0.95–1.19) | 1.15 (1.01–1.29) | 0.553 |

Antimicrobial prescription considered in total, by authorised administration route, and by priority classification; the latter also being summarised by main presenting complaint. Also included are $P$ value outputs of a series of mixed effects panel regression models, modelling prescription category against intervention group, pre- and post-intervention. Full regression model outputs are available in Supplementary Tables 2 (Antimicrobial prescription), 7 (HPCIA prescription by main presenting complaint) and 9 (other prescriptions). Multiple authorised administration route antimicrobials can be prescribed in a single consultation. Data is derived from 88,298 pre-intervention CG consultations; 107,223 pre-intervention and 81,582 post-intervention LIG consultations, and 63,366 pre-intervention and 63,301 post-intervention HIG consultations.
[a]Highest priority critically important antimicrobial.
[b]Emboldened text refers to findings which were found to be significantly different on comparison to the control group.

**Table 3 Feline antimicrobial and anti-inflammatory prescription, and euthanasia as a percentage of total feline consultations by intervention group, split into pre-intervention (August 2018–March 2019) and post-intervention (April–November 2019) phases.**

| FELINE Variable | Control group | | Light intervention group | | | Heavy intervention group | | |
|---|---|---|---|---|---|---|---|---|
| | Pre-intervention | Post-intervention | Pre-intervention | Post-intervention | P | Pre-intervention | Post-intervention | P |
| **Primary outcome** | | | | | | | | |
| HPCIA[a] | 7.44 (6.27–8.61) | 7.37 (6.02–8.72) | 7.03 (6.32–7.74) | **6.14 (5.29–6.99)**[b] | **0.030** | 8.03 (7.16–8.90) | **4.35 (3.41–5.29)** | **<0.0001**c |
| **Antimicrobial prescription—% of total consultations (95% confidence interval, CI)** | | | | | | | | |
| Total | 16.68 (15.19–18.17) | 16.62 (15.35–17.88) | 14.97 (13.46–16.49) | 14.78 (13.35–16.20) | 0.103 | 15.26 (14.34–16.18) | **14.06 (12.59–15.52)** | **0.002** |
| Systemic | 14.08 (12.60–15.55) | 13.84 (12.61–15.08) | 12.43 (11.13–13.74) | 12.00 (10.64–13.36) | 0.098 | 12.75 (11.91–13.58) | **11.31 (9.93–12.68)** | **0.002** |
| Topical | 3.03 (2.70–3.35) | 3.27 (2.94–3.60) | 2.98 (2.64–3.33) | 3.18 (2.90–3.45) | 0.297 | 2.95 (2.62–3.28) | 3.12 (2.84–3.40) | 0.073 |
| Systemic HPCIA | 7.35 (6.12–8.58) | 7.29 (5.98–8.59) | 6.93 (6.23–7.63) | **6.05 (5.19–6.91)** | **0.026** | 7.95 (7.08–8.82) | **4.25 (3.31–5.20)** | **<0.0001**d |
| **HPCIA prescription by main presenting complaint—% of relevant consultations (95% CI)** | | | | | | | | |
| Vaccination | 0.76 (0.34–1.19) | 1.04 (0.54–1.55) | 1.02 (0.51–1.52) | 1.14 (0.54–1.74) | 0.933 | 1.35 (0.35–2.36) | 0.59 (0.39–0.78) | 0.090 |
| Other healthy | 5.17 (3.42–6.93) | 6.29 (4.34–8.25) | 5.65 (4.49–6.82) | 5.06 (3.89–6.23) | 0.179 | 5.92 (4.29–7.55) | **3.60 (2.16–5.04)** | **0.001** |
| Post-operative check | 5.81 (1.88–9.75) | 5.65 (0.96–10.35) | 3.50 (2.06–4.94) | 3.42 (2.45–4.38) | 0.742 | 4.54 (3.05–6.03) | 3.11 (1.83–4.39) | 0.833 |
| Gastroenteric | 4.86 (3.19–6.53) | 4.26 (1.78–6.74) | 4.45 (2.34–6.56) | 3.18 (1.29–5.08) | 0.672 | 6.44 (4.00–8.88) | 2.27 (0.94–3.61) | 0.306 |
| Respiratory | 27.35 (20.07–34.63) | 25.05 (14.19–35.90) | 24.99 (19.21–30.77) | 17.93 (12.92–22.95) | 0.691 | 32.19 (24.15–40.23) | 13.46 (7.54–19.38) | 0.062 |
| Pruritus | 15.71 (11.25–20.17) | 14.93 (9.64–20.22) | 17.76 (13.63–21.90) | 13.73 (10.77–16.69) | 0.311 | 19.70 (14.05–25.35) | 9.57 (5.73–13.41) | 0.066 |
| Trauma | 28.86 (22.65–35.08) | 26.96 (21.66–32.26) | 27.39 (24.18–30.60) | 23.08 (17.89–28.27) | 0.202 | 29.19 (24.71–33.66) | **13.96 (9.63–18.28)** | **0.001** |
| Tumour | 13.20 (10.09–16.32) | 13.77 (9.30–18.23) | 12.03 (9.25–14.82) | 8.62 (5.86–11.38) | 0.059 | 13.24 (8.80–17.68) | 7.25 (4.10–10.39) | 0.095 |
| Kidney disease | 15.28 (7.27–23.28) | 15.10 (6.75–23.45) | 13.02 (8.34–17.69) | 10.08 (7.05–13.12) | 0.420 | 15.69 (11.67–19.71) | 9.83 (4.87–13.79) | 0.412 |
| Other unwell | 14.13 (11.79–16.47) | 12.98 (10.98–14.91) | 13.25 (11.49–15.01) | 11.58 (9.94–13.21) | 0.382 | 14.99 (13.01–16.97) | **8.13 (6.14–10.12)** | **0.001** |
| **Other prescriptions—% of total consultations (95% CI)** | | | | | | | | |
| Anti-inflammatory | 19.03 (17.12–20.93) | 19.46 (18.17–20.76) | 17.50 (16.23–18.76) | 17.81 (16.68–18.93) | 0.321 | 18.67 (17.06–20.28) | 18.71 (17.16–20.25) | 0.578 |
| Euthanasia | 2.14 (1.64–2.65) | 2.15 (1.64–2.65) | 2.29 (1.99–2.58) | 2.33 (1.97–2.68) | 0.684 | 2.13 (1.89–2.38) | 2.29 (1.89–2.70) | 0.645 |

Antimicrobial prescription considered in total, by authorised administration route, and by priority classification; the latter also being summarised by main presenting complaint. Also included are P value outputs of a series of mixed effects panel regression models, modelling prescription category against intervention group, pre- and post-intervention. Full regression model outputs are available in Supplementary Tables 3 (Antimicrobial prescription), 8 (HPCIA prescription by main presenting complaint) and 9 (other prescriptions). Multiple authorised administration route antimicrobials can be prescribed in a single consultation. Data is derived from 33,613 pre-intervention and 30,632 post-intervention CG consultations; 39,803 pre-intervention and 37,339 post-intervention LIG consultations, and 25,661 pre-intervention and 25,808 post-intervention HIG consultations.
aHighest priority critically important antimicrobial.
bEmboldened text refers to findings which were found to be significantly different on comparison to the control group.
c0.0000006.
d0.0000005.

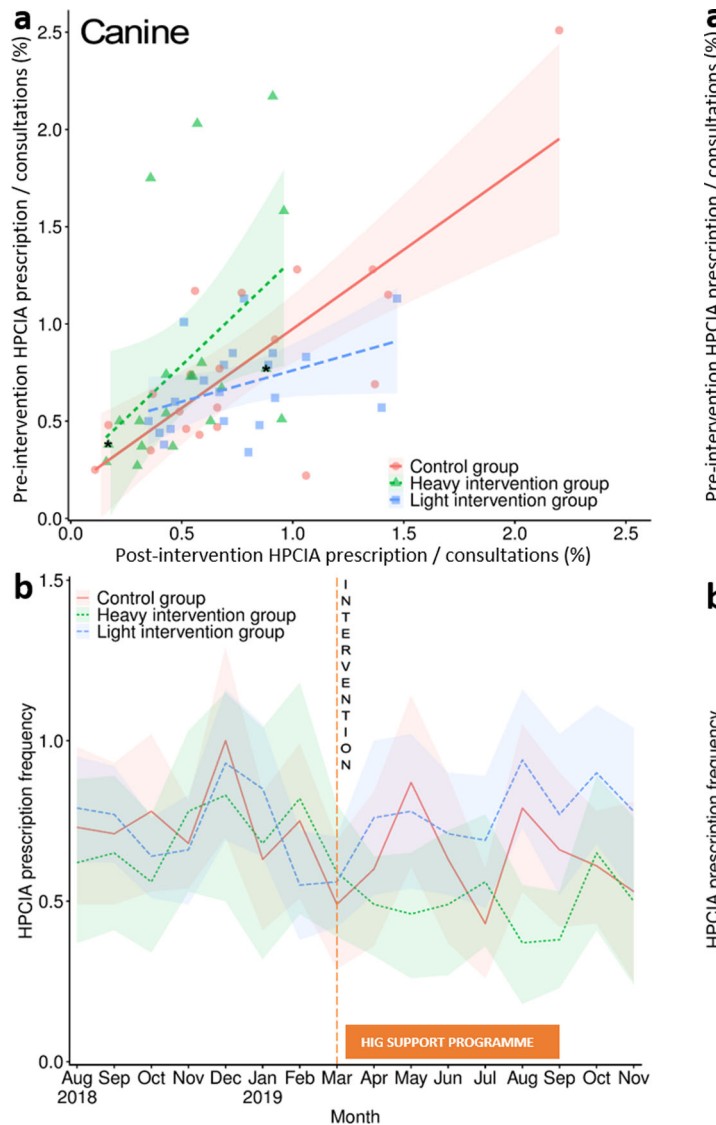

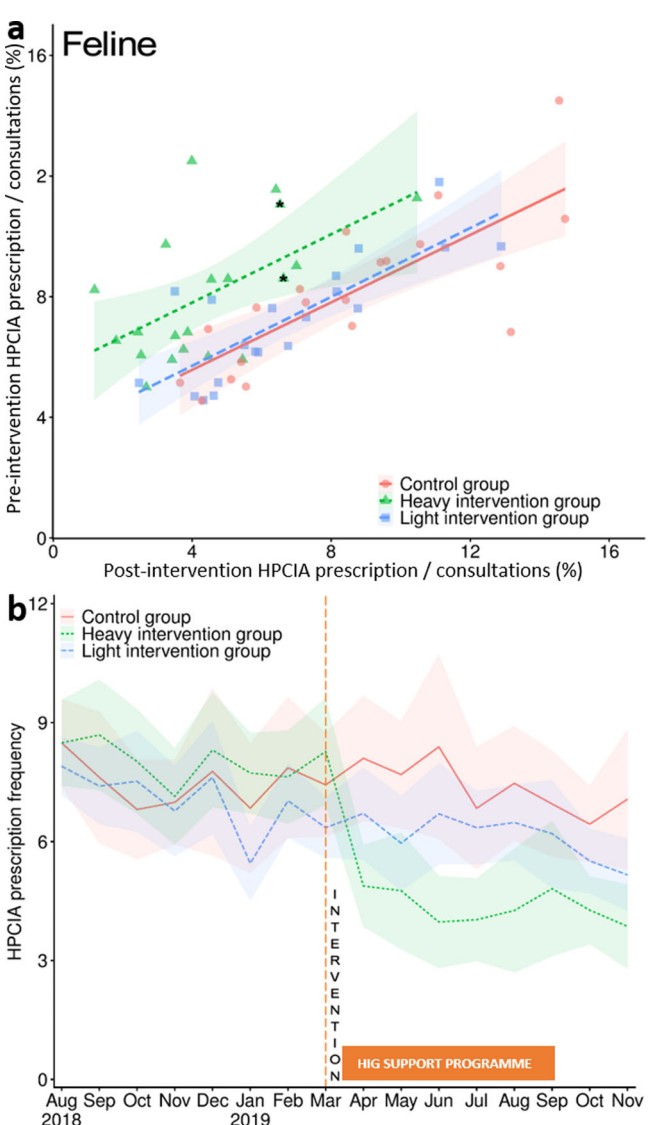

**Fig. 2 Canine highest priority critically important antimicrobial prescription frequency.** Canine highest priority critically important antimicrobial (HPCIA) prescription frequency as a percentage of total consultations, measured by (**a**) practice pre-intervention (August 2018–March 2019 inclusive) and post-intervention (April–November 2019 inclusive), and (**b**) month. Lines in plot a refer to linear regression fits, modelling intervention status by practice, with lines referring to regression fit estimates around a 95% confidence interval (shaded region). An asterisk refers to the two practices in the HIG which declined to participate in the post-benchmarking intervention reflection and education programme. Lines in plot b refer to HPCIA prescription frequency as a percentage of total consultations; the red solid line in both plots a and b refers to findings from the control group (CG); the blue hashed line refers to the light intervention group (LIG), and the green dotted line refers to the heavy intervention group (HIG). The orange dashed line in plot b shows the month at which the initial notification took place, and the orange solid box outlines the months in which the HIG could access further support if requested. Shaded regions refer to 95% confidence intervals. Data is derived from 88,298 pre-intervention and 81,582 post-intervention CG consultations; 107,223 pre-intervention and 105,518 post-intervention LIG consultations, and 63,366 pre-intervention and 63,301 post-intervention HIG consultations.

**Fig. 3 Feline highest priority critically important antimicrobial prescription frequency.** Feline highest priority critically important antimicrobial (HPCIA) prescription frequency as a percentage of total consultations, measured by (**a**) practice pre-intervention (August 2018–March 2019 inclusive) and post-intervention (April–November 2019 inclusive), and (**b**) month. Lines in plot a refer to linear regression fits, modelling intervention status by practice, with lines referring to regression fit estimates around a 95% confidence interval (shaded region). An asterisk refers to practices in the HIG which declined to participate in the post-benchmarking intervention reflection and education programme. Lines in plot b refer to HPCIA prescription frequency as a percentage of total consultations; the red solid line in both plots a and b refers to findings from the control group; the blue hashed line refers to the light intervention group, and the green dotted line refers to the heavy intervention group (HIG). The orange dashed line in panel b shows the month at which the initial notification took place, and the orange solid box outlines the months in which the HIG could access further support if requested. Shaded regions refer to 95% confidence intervals. Data is derived from 33,613 pre-intervention and 30,632 post-intervention CG consultations; 39,803 pre-intervention and 37,339 post-intervention LIG consultations, and 25,661 pre-intervention and 25,808 post-intervention HIG consultations.

**Table 4 Canine and feline HPCIA prescription as a percentage of total consultations, summarised by month between August 2018 and November 2019.**

| Month | Control group | Light intervention group | | Heavy intervention group | |
|---|---|---|---|---|---|
| | HPCIA[a] % (CI)[b] | HPCIA % (CI) | P | HPCIA % (CI) | P |
| CANINE | | | | | |
| August, 2018 (pre-intervention) | 0.73 (0.48–0.98) | 0.79 (0.62–0.95) | 0.369 | 0.62 (0.37–0.88) | 0.623 |
| September, 2018 | 0.71 (0.48–0.94) | 0.77 (0.62–0.92) | 0.452 | 0.65 (0.41–0.90) | 0.741 |
| October, 2018 | 0.78 (0.53–1.02) | 0.64 (0.51–0.76) | 0.174 | 0.56 (0.34–0.78) | 0.122 |
| November, 2018 | 0.67 (0.54–0.81) | 0.66 (0.49–0.83) | 1.000 | 0.79 (0.53–1.05) | 0.780 |
| December, 2018 | 1.00 (0.70–1.30) | 0.93 (0.69–1.16) | 0.639 | 0.83 (0.51–1.15) | 0.513 |
| January, 2019 | 0.62 (0.41–0.84) | 0.84 (0.64–1.05) | 0.501 | 0.68 (0.32–1.03) | 0.129 |
| February, 2019 | 0.75 (0.52–0.99) | 0.55 (0.38–0.72) | 0.164 | **0.82 (0.45–1.18)[c]** | **0.043** |
| March, 2019 | 0.49 (0.29–0.70) | 0.56 (0.42–0.70) | 0.986 | 0.59 (0.39–0.80) | 0.418 |
| April, 2019 (post-intervention) | 0.60 (0.36–0.85) | 0.76 (0.52–1.00) | 0.909 | 0.49 (0.33–0.64) | 0.222 |
| May, 2019 | 0.88 (0.62–1.14) | 0.78 (0.54–1.03) | 0.338 | **0.46 (0.26–0.66)** | **0.022** |
| June, 2019 | 0.63 (0.37–0.89) | 0.71 (0.52–0.90) | 0.437 | 0.49 (0.27–0.72) | 0.220 |
| July, 2019 | 0.43 (0.26–0.60) | 0.69 (0.48–0.89) | 0.536 | 0.56 (0.36–0.77) | 0.399 |
| August, 2019 | 0.79 (0.54–1.04) | 0.95 (0.74–1.16) | 0.568 | **0.37 (0.18–0.56)** | **0.005** |
| September, 2019 | 0.66 (0.42–0.90) | 0.77 (0.52–1.01) | 0.382 | **0.38 (0.23–0.53)** | **0.042** |
| October, 2019 | 0.61 (0.43–0.78) | 0.90 (0.68–1.11) | 0.335 | 0.65 (0.41–0.89) | 0.910 |
| November, 2019 | 0.53 (0.25–0.81) | 0.78 (0.52–1.04) | 0.912 | 0.50 (0.24–0.76) | 0.322 |
| FELINE | | | | | |
| August, 2018 (pre-intervention) | 8.49 (7.42–9.57) | 7.90 (7.16–8.63) | 0.337 | 8.47 (7.41–9.53) | 0.878 |
| September, 2018 | 7.60 (5.95–9.25) | 7.40 (6.45–8.35) | 0.799 | 8.70 (7.31–10.09) | 0.308 |
| October, 2018 | 6.81 (5.55–8.07) | 7.54 (6.24–8.83) | 0.857 | 8.03 (6.73–9.33) | 0.770 |
| November, 2018 | 7.01 (5.93–8.08) | 6.78 (5.66–7.90) | 0.574 | 7.14 (5.95–8.32) | 0.394 |
| December, 2018 | 7.77 (5.65–9.89) | 7.63 (6.21–9.06) | 0.626 | 8.31 (6.88–9.73) | 0.397 |
| January, 2019 | 6.83 (5.22–8.44) | 5.44 (4.51–6.37) | 0.312 | 7.74 (6.73–8.76) | 0.874 |
| February, 2019 | 7.86 (6.10–9.62) | 7.03 (6.09–7.98) | 0.060 | 7.65 (6.47–8.82) | 0.166 |
| March, 2019 | 7.42 (6.10–8.75) | 6.37 (5.58–7.15) | 0.557 | 8.29 (6.93–9.64) | 0.572 |
| April, 2019 (post-intervention) | 8.10 (6.50–9.69) | 6.71 (5.53–7.89) | 0.325 | **4.88 (3.84–5.92)** | **0.001** |
| May, 2019 | 7.68 (6.36–9.00) | 5.98 (4.76–7.20) | 0.195 | **4.77 (3.32–6.22)** | **0.001** |
| June, 2019 | 8.35 (6.00–10.70) | **6.69 (5.43–7.94)** | **0.033** | **3.97 (2.82–5.13)** | **<0.0001[d]** |
| July, 2019 | 6.86 (5.29–8.43) | 6.34 (5.43–7.25) | 0.788 | **4.05 (2.99–5.11)** | **0.005** |
| August, 2019 | 7.49 (6.03–8.95) | 6.47 (5.52–7.42) | 0.110 | **4.28 (2.72–5.83)** | **<0.0001[e]** |
| September, 2019 | 6.95 (5.61–8.30) | 6.19 (4.79–7.60) | 0.740 | **4.84 (3.15–6.52)** | **0.006** |
| October, 2019 | 6.44 (5.48–7.40) | 5.51 (4.70–6.33) | 0.265 | **4.27 (3.41–5.12)** | **0.020** |
| November, 2019 | 7.07 (5.30–8.83) | **5.16 (4.26–6.07)** | **0.019** | **3.86 (2.80–4.92)** | **0.0006** |

P value outputs of mixed effects panel regression, modelling intervention group against month are also included. Full regression model outputs are available in Supplementary Tables 4 (dogs) and 5 (cats). Data is derived from 169,880 canine and 64,245 feline CG consultations; 212,741 canine and 67,142 feline LIG consultations, and 126,667 canine and 51,469 feline HIG consultations.
[a]Highest priority critically important antimicrobial.
[b]95% Confidence interval.
[c]Emboldened text refers to findings which were found to be significantly different on comparison to the control group.
[d]0.000002.
[e]0.00004.

antimicrobial stewardship scheme, led by RCVS Knowledge (https://knowledge.rcvs.org.uk/home/), firmly demonstrating a profession-wide commitment to responsible usage of antimicrobials.

In the LIG and HIG, use of the SAVSNET antimicrobial benchmarking portal significantly increased post-notification, with increases being most apparent in the 2 months following notification, suggesting that notifications informing practitioners of their relatively "unusual" status in relation to HPCIA prescription frequency prompted enhanced portal engagement. However, while post-intervention 80% of HIG practices interacted with the portal, only 40% of LIG practices did so. Furthermore, while engagement waned to at or below pre-intervention levels within two months of notification in the LIG, interest exceeded pre-intervention levels for 6 months post-notification in the HIG. It is probable that either the additional in-depth benchmarking report provided to practices in the HIG, or post-notification offer of assistance from the hub clinical leads might have enhanced initial interest compared to the LIG. Though the relative contributions of each was not able to be elucidated here, individuals are more likely to re-evaluate existing

behaviours if modifying behaviour might bring reward, or not doing so might bring punishment[40]. Although the supportive, optional nature of the trial was emphasised throughout, requested hub clinical lead intercession might have nevertheless introduced a perception of potential reward or punishment linked with engagement with the trial. It is also possible that the letter and email sent to LIG practices was not disseminated beyond clinical directors in some cases, whereas the opportunity for practice-wide meetings held in HIG practices encouraged the engagement of more staff. It was not possible to distinguish between individual practice staff members interacting with the benchmarking portal; hence, engagement might have been limited to a single or few member(s) of each practice.

There was concern that notification of relatively high HPCIA prescription alone could prompt practice policy changes not reflective of latest clinical evidence, such changes being potentially detrimental to animal welfare or employee wellbeing. Structured reflection and education programmes have been shown to be effective at achieving sustained improvements in anti-infective prescription habits in the medical field[33], and in this

**Table 5 Canine antimicrobial prescription choice as a percentage of total antimicrobial prescriptions, presented by class and for beta-lactams, by sub-class, and by intervention group split into pre- and post-intervention time periods.**

| CANINE Antimicrobial class | Control group % (95% CI)[a] | | Light intervention group % (95% CI) | | | Heavy intervention group % (95% CI) | | |
|---|---|---|---|---|---|---|---|---|
| | Pre-intervention | Post-intervention | Pre-intervention | Post-intervention | P | Pre-intervention | Post-intervention | P |
| Aminoglycoside | 10.94 (9.00–12.88) | 5.50 (4.16–6.84) | 10.60 (8.08–13.13) | 6.17 (4.34–7.99) | 0.444 | 9.81 (7.71–11.90) | 6.04 (3.11–8.97) | 0.699 |
| Amphenicol | 7.14 (4.88–9.39) | 8.28 (6.22–10.33) | 5.38 (3.44–7.31) | 7.37 (5.23–9.51) | 0.618 | 6.60 (4.21–8.99) | 8.03 (4.85–11.21) | 0.281 |
| Beta-lactam[b] | 41.71 (37.35–46.07) | 44.84 (41.34–48.34) | 43.48 (40.85–46.11) | 45.15 (41.97–48.32) | 0.259 | 41.87 (39.20–44.53) | 41.83 (38.81–44.86) | 0.204 |
| *Amoxicillin* | *5.27 (0.58–9.97)* | *4.00 (0.11–7.89)* | *4.58 (2.21–6.94)* | *4.09 (1.84–6.33)* | *0.684* | *6.83 (1.49–12.18)* | *5.36 (1.54–9.18)* | *0.849* |
| *Clavulanic acid potentiated amoxicillin* | *73.80 (69.84–77.76)* | *73.93 (69.43–78.43)* | *73.31 (68.49–78.14)* | *76.84 (71.08–82.60)* | *0.178* | *74.46 (69.46–79.46)* | *78.44 (72.21–84.68)* | *0.242* |
| *1st generation cephalosporin* | *18.85 (14.87–22.83)* | *20.73 (16.50–24.95)* | *20.43 (15.65–25.22)* | *17.55 (12.70–22.40)* | *0.711* | *16.13 (13.14–19.11)* | *15.17 (11.23–19.11)* | *0.125* |
| *2nd generation cephalosporin* | *0.10 (0.00–0.19)* | *0.07 (0.01–0.13)* | *0.06 (0.00–0.13)* | *0.05 (0.00–0.12)* | *1.000* | *0.29 (0.03–0.54)* | *0.02 (0.00–0.06)* | *0.482* |
| *3rd generation cephalosporin [c]* | *1.71 (1.13–2.30)* | *1.10 (0.65–1.54)* | *1.31 (0.83–1.80)* | *1.35 (0.85–1.84)* | *0.165* | *2.24 (1.61–2.86)* | *0.97 (0.50–1.45)* | *0.355* |
| *Penicillin* | *–* | *–* | *–* | *0.01 (0.00–0.03)* | *0.296* | *0.02 (0.00–0.05)* | *–* | *0.117* |
| *Other beta-lactams* | *0.38 (0.00–0.92)* | *0.09 (0.00–0.24)* | *0.30 (0.00–0.62)* | *0.15 (0.00–0.36)* | *0.644* | *0.09 (0.00–0.26)* | *–* | *0.729* |
| Fluoroquinolone[c] | 2.75 (2.17–3.33) | 2.92 (2.18–3.66) | 2.97 (2.32–3.62) | 3.53 (2.57–4.49) | 0.086 | 2.50 (1.51–3.49) | 2.54 (1.88–3.21) | 0.253 |
| Fusidic acid | 18.03 (16.35–19.71) | 13.39 (11.78–15.00) | 19.40 (18.24–20.56) | 15.03 (13.06–17.01) | 0.209 | 19.97 (18.74–21.19) | 18.02 (15.82–20.21) | 0.862 |
| Lincosamide | 3.96 (2.76–5.17) | 3.89 (2.69–5.09) | 2.73 (2.06–3.40) | 2.20 (1.61–2.79) | 0.397 | 3.55 (2.32–4.78) | 3.20 (2.35–4.04) | 0.182 |
| Macrolide[c] | 0.03 (0.01–0.06) | 0.03 (0.00–0.05) | 0.04 (0.01–0.08) | 0.03 (0.01–0.06) | 0.423 | 0.03 (0.00–0.06) | 0.02 (0.00–0.04) | 0.593 |
| Nitroimidazole | 6.83 (4.96–8.70) | 7.46 (5.83–9.09) | 5.52 (3.73–7.31) | 6.13 (4.53–7.72) | 0.824 | 6.21 (4.27–8.15) | 5.65 (4.28–8.83) | 0.209 |
| Nitroimidazole-macrolide | 0.20 (0.05–0.34) | 0.19 (0.07–0.32) | 0.29 (0.06–0.51) | 0.34 (0.08–0.60) | 0.280 | 0.44 (0.00–1.10) | 0.21 (0.00–0.46) | 0.813 |
| Other antimicrobials | 7.12 (5.29–8.95) | 12.56 (10.68–14.44) | 8.86 (6.71–11.00) | 13.45 (11.28–15.61) | 0.137 | 8.25 (7.12–9.38) | 12.76 (10.88–14.65) | 0.253 |
| Sulphonamide | 0.05 (0.00–0.11) | – | 0.02 (0.00–0.05) | – | 1.000 | 0.06 (0.00–0.13) | 0.05 (0.00–0.12) | 0.232 |
| Tetracycline | 1.25 (0.81–1.70) | 0.99 (0.41–1.56) | 0.71 (0.46–0.95) | 0.61 (0.38–0.84) | 0.612 | 0.69 (0.38–1.00) | 0.81 (0.45–1.17) | 0.317 |

Also included are P value outputs of a series of mixed effects panel regression models, modelling antimicrobial class or sub-class against intervention group, pre- and post-intervention. Full regression model outputs are available in Supplementary Table 10. Data is derived from 18,599 pre-intervention and 15,966 post-intervention CG antimicrobial prescriptions; 22,678 pre-intervention and 20,827 post-intervention LIG antimicrobial prescriptions, and 13,199 pre-intervention and 11,024 post-intervention HIG antimicrobial prescriptions.
[a]95% Confidence interval.
[b]Findings presented in italics are expressed as a percentage of total beta-lactam prescription within intervention groups i.e., in the control group pre-intervention, amoxicillin comprised 5.3% of beta-lactam prescriptions.
[c]Highest Priority Critically Important Antimicrobial.

**Table 6 Feline antimicrobial prescription choice as a percentage of total antimicrobial prescriptions, presented by class and for beta-lactams, by sub-class, and by intervention group split into pre- and post-intervention time periods.**

| FELINE Antimicrobial class | Control group % (95% CI)[a] | | Light intervention group % (95% CI) | | | Heavy intervention group % (95% CI) | | |
|---|---|---|---|---|---|---|---|---|
| | Pre-intervention | Post-intervention | Pre-intervention | Post-intervention | P | Pre-intervention | Post-intervention | P |
| Aminoglycoside | 3.36 (2.54–4.18) | 1.89 (0.97–2.81) | 3.95 (2.77–5.13) | 2.31 (1.38–3.23) | 0.515 | 3.62 (2.74–4.49) | 2.25 (1.55–2.95) | 0.837 |
| Amphenicol | 2.50 (0.80–4.20) | 3.42 (1.76–5.08) | 1.91 (1.48–2.35) | 1.91 (1.44–2.40) | 0.318 | 1.92 (1.07–2.77) | 3.05 (1.87–4.22) | 0.363 |
| Beta-lactam[b] | 74.53 (69.92–79.14) | 74.82 (71.43–78.20) | 73.08 (71.04–75.12) | 73.49 (70.69–76.29) | 0.742 | 70.63 (67.22–74.05) | 71.29 (68.72–73.86) | 0.277 |
| Amoxicillin | 9.11 (1.96–16.25) | 6.76 (1.20–12.32) | 6.95 (2.49–11.41) | 6.73 (2.31–11.16) | 0.931 | 6.84 (3.70–9.99) | 9.33 (4.86–13.80) | 0.957 |
| Clavulanic acid potentiated amoxicillin | 37.80 (28.44–47.16) | 40.12 (31.32–48.93) | 37.07 (30.73–43.40) | 43.17 (36.18–50.15) | 0.636 | 29.14 (23.72–34.57) | 52.59 (43.32–61.87) | 0.776 |
| 1st generation cephalosporin | 0.72 (0.43–1.00) | 0.84 (0.47–1.21) | 1.31 (0.33–2.29) | 1.47 (0.54–2.40) | 0.257 | 0.96 (0.19–1.72) | 1.59 (0.42–2.75) | 0.715 |
| 2nd generation cephalosporin | 0.02 (0.00–0.06) | 0.05 (0.00–0.14) | 0.04 (0.00–0.13) | 0.02 (0.00–0.07) | 0.612 | – | 0.07 (0.00–0.16) | 1.000 |
| 3rd generation cephalosporin[c] | 52.45 (42.80–62.09) | 52.09 (42.00–62.17) | 54.57 (45.64–63.51) | 48.72 (41.97–55.48) | 0.908 | 63.11 (57.45–68.77) | 36.43 (29.10–43.75) | 0.057 |
| Penicillin | 0.06 (0.00–0.15) | 0.10 (0.00–0.32) | 0.02 (0.00–0.05) | – | 0.182 | 0.03 (0.00–0.09) | – | 0.182 |
| Other beta-lactams | – | – | – | – | – | – | – | – |
| Fluoroquinolone[c] | 0.96 (0.56–1.36) | 1.16 (0.88–1.44) | 1.55 (1.10–2.01) | 1.29 (0.92–1.66) | 0.708 | 3.14 (1.00–5.29) | 1.47 (0.87–2.07) | 0.894 |
| Fusidic acid | 10.50 (9.00–12.01) | 9.19 (7.65–10.73) | 11.53 (10.36–12.70) | 11.58 (9.92–13.24) | 0.217 | 11.79 (10.68–12.91) | 11.82 (10.03–13.61) | 0.880 |
| Lincosamide | 3.06 (1.61–4.51) | 2.91 (1.67–4.16) | 3.04 (1.14–4.93) | 2.93 (1.02–4.83) | 0.808 | 2.81 (1.88–3.75) | 3.37 (2.23–4.52) | 0.808 |
| Macrolide[c] | – | 0.02 (0.00–0.06) | – | – | 0.317 | 0.02 (0.00–0.07) | 0.03 (0.00–0.08) | 1.000 |
| Nitroimidazole | 1.11 (0.57–1.66) | 1.08 (0.60–1.56) | 0.59 (0.31–0.87) | 0.75 (0.50–1.00) | 0.509 | 1.44 (0.91–1.97) | 1.21 (0.75–1.68) | 0.637 |
| Nitroimidazole-macrolide | 0.32 (0.01–0.63) | 0.30 (0.10–0.50) | 0.12 (0.02–0.21) | 0.11 (0.02–0.20) | 0.347 | 0.31 (0.00–0.66) | 0.18 (0.00–0.40) | 0.347 |
| Other antimicrobials | 2.37 (1.60–3.13) | 4.11 (3.09–5.13) | 2.99 (2.45–3.53) | 4.62 (3.82–5.42) | 0.300 | 2.99 (2.39–3.59) | 3.80 (2.98–4.62) | 0.198 |
| Sulphonamide | – | 0.01 (0.00–0.04) | 0.01 (0.00–0.04) | – | – | – | – | – |
| Tetracycline | 1.35 (0.41–2.28) | 1.10 (0.38–1.83) | 1.22 (0.81–1.63) | 1.02 (0.61–1.42) | 0.961 | 1.31 (0.29–2.33) | 1.52 (0.64–2.40) | 0.942 |

Also included are P value outputs of a series of mixed effects panel regression models, modelling antimicrobial class or sub-class against intervention group, pre- and post-intervention. Full regression model outputs are available in Supplementary Table 11. Data is derived from 6296 pre-intervention and 5575 post-intervention CG antimicrobial prescriptions; 6762 pre-intervention and 6183 post-intervention LIG antimicrobial prescriptions, and 4351 pre-intervention and 4107 post-intervention HIG antimicrobial prescriptions.
Also included are P value outputs of a series of mixed effects panel regression models, modelling antimicrobial class or sub-class against intervention group, pre- and post-intervention. Full regression model outputs are available in Supplementary Table 11. Data is derived from 6296 pre-intervention and 5575 post-intervention CG antimicrobial prescriptions; 6762 pre-intervention and 6183 post-intervention LIG antimicrobial prescriptions, and 4351 pre-intervention and 4107 post-intervention HIG antimicrobial prescriptions.
[a]95% Confidence interval.
[b]Highest Priority Critically Important Antimicrobial.
[c]Findings presented in italics are expressed as a percentage of total beta-lactam prescription within intervention groups i.e., in the control group pre-intervention, amoxicillin comprised 9.1% of beta-lactam prescriptions.

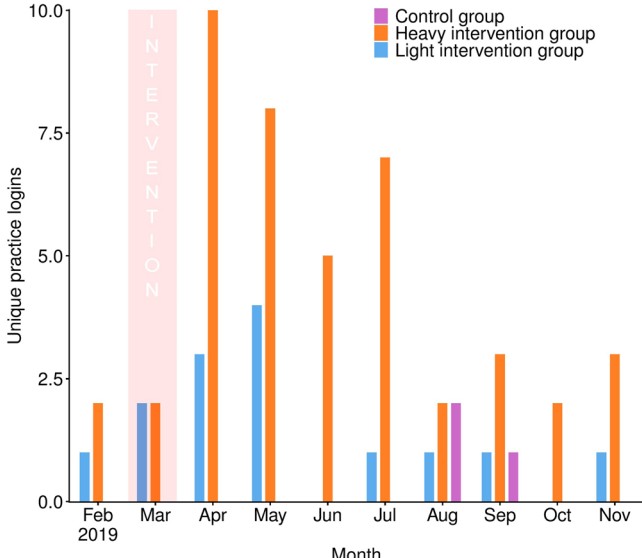

**Fig. 4 SAVSNET antimicrobial prescription benchmarking portal engagement.** Number of practices logging into the SAVSNET antimicrobial prescription benchmarking portal by month (February–November 2019 inclusive) and intervention group. The purple boxes refer to findings from the control group (n = 20 practices); the blue boxes refer to the light intervention group (n = 20 practices), and the green orange boxes refer to the heavy intervention group (HIG, n = 20 practices). The pink shaded region shows the month at which the initial notification took place.

trial we compared both a light (LIG) and heavy (HIG) reflection and educational intervention. While significant reductions in HPCIA prescription frequency were seen in both species in the HIG, significant decreases were only observed in the LIG for cats. Across the veterinary sector impressive reductions have been achieved over the past five years, especially in pigs and poultry[41,42], utilising a variety of statutory[43] and voluntary[44] improvement measures. Though over this time reductions in antimicrobial prescription frequency in companion animals have been noted[10,12,14], HPCIA use has remained an issue, particularly in cats[10,11,17]. Unlike other veterinary sectors, no statutory policies have been introduced to prompt improvements in antimicrobial prescription in companion animals. It is unknown what impact such enforced measures might have on animal welfare, and thus we consider findings presented here to be an encouraging sign that practitioners might be willing to voluntarily engage with improvement efforts, potentially negating need for firmer regulatory approaches.

This study further demonstrated the relative ease by which EHRs can be utilised to both identify participants and monitor key outcomes in near real-time. Such efficiency advantages have been previously outlined in medical research, enabling rapid scaling of interventions to instigate national quality improvement[34]. Only comparatively recently have EHRs become available for research and surveillance in the veterinary sector[45], and we believe this work serves as a promising demonstration of what could be achieved using EHR data-led approaches, expanding beyond practitioner-focused interventions to those encompassing owners, or pragmatic efficacy assessment of surgical and pharmaceutical interventions in routine practice, for example.

Though the intervention package provided in this trial represented a comprehensive approach to encouraging evidence-based behaviour change, it did limit our ability to determine which individual components might have been of greatest impact. Interestingly, though both LIG and HIG practices received a high feline HPCIA prescription frequency notification at the beginning

of the trial, reductions in the LIG were more modest compared to the HIG, where all HIG practices, including both HIG practices that refused engagement with the reflection and education programme, reported decreases post-notification. Though variation in scale of reduction was evident, circumstantially, these findings might suggest that hub clinical lead involvement was a motivational factor in prompting behavioural change. In either case, the aforementioned refusals do indicate a limiting factor in intervention scalability to a wider audience. Furthermore, this trial benefited from utilising existing quality improvement management structures within a single large practice group for the HIG. Thus, there remains a question as to whether this intervention would be feasible amongst other practices, including those that are relatively infrequent HPCIA prescribers, over a longer period of time than the eight months observed here.

Though the scale of feline HPCIA prescription frequency reduction in the LIG was approximately half that of the HIG, LIG interventions would arguably be more amenable to national deployment. Ensuring widespread efficient dissemination, even if such efforts produce a smaller overall effect, might therefore prove preferable compared to high intensity, smaller scale action. However, there are many unique practice management systems utilised in the UK, and securing SAVSNET software compliance with all systems presents somewhat of a barrier to wider participation in either intervention approach at the current time. While it was encouraging to see no immediate reversion to pre-intervention prescribing rates in the two un-supported months where hub clinical leads did not interact with HIG practices, we regard this as insufficient evidence to demonstrate "sustained" improvements at this time, and would recommend longer follow-up periods in the future.

Some MPCs in cats and dogs in the HIG were associated with significant HPCIA prescription frequency reductions, some of which (e.g., trauma) were previously associated with frequent HPCIA prescription[10,24], despite often lacking clear clinical justification for their prescription[24]. These findings suggest a generalised culture change not necessarily being restricted to reflection on individual disease presentations. This result is supported by significant reductions in systemic and overall antimicrobial prescription frequency in both species. In HIG dogs, these wider reductions were greater than that contributed by HPCIA reductions alone, suggesting that the trial had a wider impact on discouraging antimicrobial prescription more generally. However, in HIG cats the opposite was seen; HPCIA reductions were greater than overall decreases, suggesting a tendency of some practitioners to move from prescribing a HPCIA to prescribing another non-HPCIA antimicrobial, instead of avoiding prescription altogether.

Whilst no prescription choice comparisons were significantly different, 3rd generation cephalosporin feline prescription does appear to have decreased to a degree in the HIG, while clavulanic acid potentiated amoxicillin prescription increased, suggesting a preferred alternative to 3rd generation cephalosporins. Clavulanic acid potentiated amoxicillin is an authorised, widely used antimicrobial in veterinary practice[10]. However, like 3rd generation cephalosporins, use of clavulanic acid potentiated amoxicillin has also been associated with resistance development[4], is considered of 'high' importance by the WHO[16], and accordingly is only infrequently prescribed to humans in the UK[46]. It is therefore possible that in seeking to reduce risk for development of resistance in response to HPCIA prescription that this trial has inadvertently enhanced or at least sustained resistance development risk for broader beta-lactam resistance. Thus, future stewardship efforts will need to expand scope beyond HPCIAs to also consider how to promote responsible use of all antimicrobials, and indeed other medicines too.

For instance, we have previously reported a tendency for antimicrobials and anti-inflammatories to be prescribed at the same time, despite perhaps limited clinical evidence to suggest necessity for both pharmaceutical agents[15]. However, we have also noted a recent reversing trend for respiratory disease whereby antimicrobial prescription frequency has decreased whilst anti-inflammatory prescription frequency has increased[12]. These findings perhaps reflect increasing recognition of non-bacterial mediators for respiratory disease[47], or increased attention to prescribing guidance[18]. It was interesting to note that no significant variation in anti-inflammatory prescription was observed here, perhaps demonstrating more generalised "de-coupling" of anti-inflammatory and antimicrobial prescription. Though measuring frequency of use represents a relatively simple method for demonstrating change, reduced use is not necessarily representative of more responsible use. Hence, it is probable that more nuanced methods for ascertaining whether a pharmaceutical agent has been prescribed appropriately will be needed, such as consideration of specific clinical signs and/or diagnostic test findings at point of prescription. As part of these developments, we would advocate increased attention on use of other pharmaceutical agents that might form effective alternatives to antimicrobial prescription, whilst also satisfying the recognised need of a practitioner to provide a clear demonstration of action via provision of a therapeutic product to the client[19].

Though prescribers retained full autonomy to prescribe what they considered best for the animal under their care, we incorporated euthanasia frequency as a relatively crude measure of any increase in adverse health effects associated with change in prescription decision-making prompted by this trial. While no significant increases were observed in either intervention group, compared to the CG, we recognise that this method lacks sensitivity, not taking into account a range of potential sub-optimal outcomes that might compromise animal welfare. Though effectively and efficiently quantifying such adverse effects from EHRs at the scale required for this trial presents a significant challenge[48], we recommend further development of text-mining and statistical methodologies to explore such nuances for subsequent trials.

Use of bacterial infection-associated diagnostic tests was not significantly altered in this trial. Low frequency of use of such tests has been identified as a barrier to effective stewardship[49], likely reflecting low confidence in their ability to provide timely, useful clinical insights[50]. Indeed, across all groups test orders were low, indicating a preference for empirical antimicrobial prescription throughout the trial. Used correctly, these tests do play an important role in correctly managing a patient[51]; however, there is clearly more work needed to convince practitioners —and owners—of the benefits of regularly pursuing these diagnostic routes. That said, during this trial a lack of equipment and training for cytological examinations within practices was identified, which resulted in wide-scale equipment and training provision (unpublished observations). Thus, there is hope that significant impact will be generated beyond the confines of this trial.

To conclude, in this trial we outlined a data-led benchmarking, reflection and education antimicrobial stewardship framework that successfully reduced HPCIA, systemic and overall antimicrobial prescription frequency in dogs and cats in practices belonging to a single large practice group. However, whilst initially encouraging, further work is required to understand the relative impact of different antimicrobials on conferring clinically meaningful resistance, and how to incentivise increased use of diagnostic testing in preference to empirical antimicrobial prescription. This work provides a robust evidence base for future antimicrobial stewardship interventions in companion animal practice, and findings are now being used to inform development of a national stewardship scheme, in collaboration with RCVS Knowledge and CVS Group (UK) Limited.

## Methods

**Data collection**. This trial used data collected by the Small Animal Veterinary Surveillance Network (SAVSNET) project, which harnesses voluntarily provided EHR from booked consultations in a sentinel network of UK veterinary practices, held in a SQL database and queried via Microsoft SQL Server Management Studio 18. Each EHR includes anonymised information pertaining to the animal and owner, the clinical narrative, and products dispensed during such consultations. Antimicrobial prescription was identified via reference to products dispensed, and classified into systemic (oral or injectable) or topical (topical, aural, ocular) authorised administration routes using a semi-automated rule-based text-mining method, using the Veterinary Medicine Directorate's Product Information Database and the electronic Medicines Compendium (Datapharm Communications) as a guide for veterinary and human-authorised products, respectively[10]. Fluoroquinolones, macrolides and 3rd generation cephalosporins were considered HPCIAs[16]. Every consultation was further classified by the attending veterinary professional into one of ten main presenting complaints (MPCs), indicating the main reason the animal was presented to the veterinary practice[10].

In addition, CVS Group Ltd. provided data relating to staff numbers per practice (in full-time equivalents, FTE), and the number of cytological or bacterial culture and susceptibility tests ordered by practices included in this trial. SAVSNET holds ethical approval to collect EHR data from the University of Liverpool (ethical approval reference: RETH000964); additional approval was granted to encompass interventions and data collection specific to this trial from the Universities' Veterinary Research Ethics Committee (ethical approval reference: VREC745).

**Study practice selection**. This three-armed randomised controlled trial (RCT) initially utilised EHRs voluntarily supplied by 157 UK veterinary practices (385 sites/branches) belonging to CVS Group Ltd., that had been participating in SAVSNET between 1 August 2018 and 15 January 2019. Pre-intervention, practice-level median HPCIA prescription as a percentage of total canine ($n = 409,279$) and feline ($n = 164,827$) consultations were 0.5% [range 0.0–2.3] and 5.3% [range 0.0–13.9], respectively. Practices were eligible for trial inclusion if they were within the 40% most frequent total HPCIA (including systemic and topical formulations) prescribing bracket as a percentage of total consultations in both dogs (in excess of 0.5% of canine consultations) and cats (in excess of 5.4% of feline consultations) ($n$ practices = 43). Practices where either cats ($n$ practices = 17) or dogs ($n$ practices = 8) were placed in the 40% most frequent total HPCIA prescribing bracket, with the opposing species being placed in the 40-60% most frequent total HPCIA prescribing bracket [dogs, range 0.4-0.5%; cats, 4.5-5.3%], were also considered eligible.

The primary outcome was HPCIA prescription frequency post-intervention, compared to the CG. A sample size estimation indicated that to detect a 10% relative decrease in the primary outcome (standard deviation = 10%, power = 80% and $\alpha$ = 0.05), 17 practices would be required in each group. Hence, of the 68 initially eligible practices, 60 practices were randomly and evenly allocated into three intervention groups: the control group (CG, $n$ practices=20, sites=40), low group (LIG, $n$ practices=20, sites=57) or high group (HIG, $n$ practices=20, sites=51) by block random allocation, utilising the 'complete random allocation' function available through the 'randomizr' R package version 0.20.0 (Fig. 1)[52]. Practice allocation was completed by DS.

**Intervention**. For the LIG and HIG, the trial consisted of two phases: (1) an initial notification of above median HPCIA prescription frequency status, followed by (2) a voluntary reflection and education programme, with intensity varying by intervention group.

On 28 March 2019, LIG practices received a posted letter and email (both addressed to the clinical director of that practice) stating their above average HPCIA-prescribing status (Supplementary Note 1). They also received a copy of the practice group's antimicrobial prescribing policy (based on current prescribing guidance)[18], a reminder and interpretive guidance for an online anonymised antimicrobial prescription benchmarking portal already freely available to all SAVSNET-participating practices (Supplementary Note 2)[30], and access to AMR educational videos (can be viewed here: https://savsnetvet.liverpool.ac.uk/savsnetamr/iv?id=61). Practices could opt-out of or access these as many times as they wished, and clinical directors were able to distribute these materials amongst staff members as they saw fit; they were not prompted to do so.

On 28 and 29 March 2019, HIG practices also received a posted letter and email (both addressed to the clinical director of that practice) stating their above average HPCIA-prescribing status. This letter included all LIG materials, and further included an in-depth benchmarking report (Supplementary Note 3) and explanatory video, in addition to the videos available to LIG practices (can be viewed here: https://savsnetvet.liverpool.ac.uk/savsnetamr/iv?id=62). Practices were further invited, via their clinical director, to participate in a reflection and education programme delivered by a "hub clinical lead" (a member of the senior clinical team recruited internally by CVS Group Ltd. based on clinical experience).

This programme consisted of an initial in-person review between a hub clinical lead and the clinical director and/or head veterinary surgeon of the practice; presence of additional members of staff were left to the discretion of each practice. Reviews were structured around a series of questions aimed at identifying factors (e.g., staffing, equipment, attitudes) that might have contributed to their relatively high HPCIA prescription frequency status (full question list available: Supplementary Note 4). Staff were able to suggest their own action points, and were encouraged to hold a separate practice-wide meeting to discuss findings.

Hub clinical leads checked in by email with HIG practices on a monthly basis following initial intervention, reminding practices to arrange the aforementioned practice-wide meeting if not already done so. Practices could also hold additional follow-up reviews with the hub clinical lead if they wished. A final hub clinical lead review near conclusion of the study was also requested, each utilising a development of the initial contributory factors question sheet (Supplementary Note 4).

Though individual LIG and HIG practices were aware of their involvement in a trial, they were not informed of which other practices were involved, nor interventions being performed in opposing intervention groups. The CG received no intervention beyond sustained access to the antimicrobial prescription benchmarking portal through SAVSNET, and remained unaware of their involvement in this trial. Presence and frequency of SAVSNET prescription benchmarking portal access was monitored throughout the trial for all practices included in this trial. It was not practical to blind study team members to group allocation.

**Outcomes**. Post-intervention monitoring was carried out between 1 April 2019 and 30 November 2019 (inclusive). The primary outcome measure was post-intervention canine or feline total HPCIA (including systemic and topical formulations) prescription frequency as a percentage of total consultations. Secondary outcome measures included post-intervention total, systemic and topical antimicrobials; systemic HPCIA prescription frequency; total HPCIA prescription frequency by month; total HPCIA prescription frequency by MPC; relative antimicrobial class prescription frequency; anti-inflammatory prescription, and euthanasia frequency. Post-intervention cytological or bacterial culture and susceptibility test order numbers, and interactions with the online antimicrobial prescription benchmarking portal were also summarised.

**Statistical analyses**. The statistical programme "R" was used for all analyses (version 4.0.3). Descriptive proportions and 95% confidence intervals were adjusted for clustering within practices (bootstrap method, $n = 5,000$ samples)[53]. Baseline characteristic variation between practices within each intervention group were explored via Kruskal-Wallis tests.

Mixed effects panel regression models, modelling practice as the random effect, were used for making intervention group comparisons, utilising the R package 'plm' (version 3.2-5)[54]. For each of the outcome measures described, practice-level pre-intervention (August 2018 – March 2019) and post-intervention (April – November 2019) values were compared with intervention group (modelled as interacting variables). For total HPCIA prescription frequency, intervention group was also compared across all pre- and post-intervention months. The CG was used as the reference category for all analyses. Intervention groups were also individually modelled via an orthogonal polynomial method to analyse temporal trend, considering up to quartic fits (Supplementary Table 1)[55]. Model assumptions were met for all models; goodness of fit was assessed via comparison against a null model. The R package "ggplot2" (version 3.3.2)[56] was used for all visualisations, with the 'lm' function being used to produce linear regression lines in Figs. 2 and 3.

Analyses were performed on an intention-to-treat basis, as EHR data were available for all practices regardless of participation opt-out status. Number of practices within each group interacting with SAVSNET's antimicrobial prescription benchmarking portal were compared pre- and post-intervention via a two-sided Fisher's Exact Test; post-hoc pairwise Fisher's test comparisons were also completed utilising the 'pairwiseNominalIndependence' function available via the R package "rcompanion" (version 2.3.26)[57]. The trial was completed according to CONSORT guidelines[58], and statistical significance was defined as $P < 0.05$ throughout.

**Reporting summary**. Further information on research design is available in the Nature Research Reporting Summary linked to this article.

## Data availability

Source data are provided with this paper. Other data that support the findings of this study are available from the corresponding author (D.A.S.) on reasonable request. Some of the data are not publicly available due to them containing information that could compromise research participant privacy. Source data are provided with this paper.

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

## Acknowledgements

SAVSNET is grateful for the support and major funding of BBSRC, the Dogs Trust, and BSAVA. CVS Group (UK) Limited provided support in-kind for this research via creation and postage of physical materials, and through secondment of existing staff onto this project. We particularly thank Marcus Evans, Richard Killen and Joe Williams. We also acknowledge the significant time investment of the hub clinical leads (Kate Allgood, Lisa Baker, Sinead Bennett, Rachel Clay, Ryan Davis, Elizabeth McLennan-Green, Mark Moreton and Nicola Rolph) involved in implementing many of the HIG interventions. More broadly, we thank data providers in veterinary practice (VetSolutions, Teleos, CVS and independent practitioners) and participating veterinary diagnostic laboratories (Axiom Veterinary Laboratories, Batt Laboratories, BioBest, Idexx, NationWide Laboratories Microbiology Diagnostics Laboratory at the University of Liverpool, the Department of Pathology and Infectious Diseases at the University of Surrey, and the Veterinary Pathology Group). Finally, we are grateful for the help and support provided by SAVSNET team administrator Susan Bolan.

## Author contributions

D.A.S. and G.L.P. conceived of and devised the trial design and prepared trial materials, including intervention letters, educational videos and benchmarking reports. D.A.S. collated and prepared data, conducted analysis and wrote the manuscript, with supervision from G.L.P.. P.J.M.N. and A.D.R. supported trial design, and provided oversight of manuscript drafts. B.B. initially recruited all trial practices onto the SAVSNET project, and provided communications support between SAVSNET and trial practices throughout. S.S. provided database support, created online platforms for educational videos to be viewed, and devised the practice antimicrobial benchmarking portal monitoring capability. A.R. assisted trial design, acted as a liaison between SAVSNET and CVS Group Ltd., and trained HCLs in how to correctly implement devised interventions.

## Competing interests

The authors declare no competing interests.
