## [Peer Review File · Nature Communications]

REVIEWER COMMENTS

Reviewer #1 (Remarks to the Author):

General comments

This is a well written manuscript and valuable contribution to the literature on AM stewardship. Overall it is clear and presents the results well. Some general comments include –

1. Interventions - these could have been described in more detail given these are central to the study. Some parts were but more details on the clinical lead review process and staff meeting processes would have been valuable.
2. Post-intervention duration of effect– this cut-off indicates usage after the receipt of a letter for the two intervention groups with further implementations undertaken over subsequent months. This would appear to impact on the actual duration of interventions and period of effect after interventions have been completed. Some comment is made that the interventions were short term to 6 months. It would be beneficial to expand this in the discussion further as the interventions themselves extended over 1-3 months based on the results reported and though most practices would have had the major interventions near the onset of this 6 months, for the HG group – many may not have had the practice meetings and ongoing clinical reviews for some time after the intervention start date and as such the follow up 'residual' period could be quite a bit shorter in some practice. This might limit extrapolating too far beyond the actual intervention period of effect.
2. Pre to post comparisons. The main statistical comparisons appear to be with the 3 treatment groups for the post-intervention AM usage as stated. It would have been interesting particularly for the main outcomes to compare to pre usage – not withstanding seasonal issues – was this an element of the statistical modelling? i.e. Changes over time in addition to between group change? This further element of the analysis could have been discussed in the discussion - or at least why it was not considered necessary.

Specific comments.

Abstract – page 2, line 12 clarify – presumably the EHR were used to identify high HPCIA prescribing practices which were then randomly allocated? (rather than the data used for allocation to 1 of the 3 groups)

Line 14 – confirm control group received no direct interaction? This could be here also in the abstract.

Line 18 – ideally include p values here. Clarify how also 'maintenance' of effect was assessed this may be harder to justify.

Line 20 – significant reduction or variation – clarify?

Conclusions- these could be more specific as to what interventions appear to be recommended.

Introduction

Line 41 consider also Tompson PVM 2020 re qualitative approaches.

Method

Line 70. Consider clarifying that data related to animal and owner were de-identified, or similar.

Line 93 – clarify if hpcia practice level median usage was reported, why only 66 practices were eligible? It would appear half of all 157 practices would be above the median practice level usage?

Line 101 – clarify was this 'complete random allocation' simple random allocation, block random sampling etc?

Line 106 onwards. The details re the interventions could be expanded: Who received the intervention practice email/letter and how was that planned to be communicated to vets within the practice.

Line 125 – it would be helpful to provide more details of how clinical leads were selected within the CVS team, of what the initial reviews by the clinical lead consisted of as well as how and to who the clinical lead performed their followup 'check ins'. Some more details could be provided in the main manuscript.

Outcomes.

Line 142 – it would appear the primary outcome was percent of total consults with Hpcia.

Reference to comparison to cg would appear unrequired here and could go in the statistical section.

Statistical analysis –

It would be useful to clarify that model assumptions and diagnostics/measures of fit were assessed and met (the latter in the results).

Were effects over time modelled relative to the time baseline t0 also? This was not clear. Could these have been considered within the modelling approach?

Some incidence rates and ratios are reported – it would be good to clarify their calculation in the methods.

Were baseline characteristics compared statistically? This would help support similarity of groups at baseline.

Within the results some further linear regression methods appear to have been performed? These could be mentioned within the methods.

Results

Participation, interventions and baseline data

Line 174 – though only 2 centres that did not take part, it would be interesting to know whether they were at an extreme of size / vet numbers – perhaps worth a comment here?

Line 175 – consider comparing statistically the groups on key baseline measures – this would support further that they appeared similar.

Line 178 – pre-intervention - it might be helpful to confirm these were not statistically different at baseline?

More details of the response rates and participation in practice meetings also would have been helpful – though this would be in the results.

Line 180 – here you refer to LG, HG rather than LiG HiG – ideally make the same throughout.

Line 195 – it would be useful to confirm these practice decreases were statistically significant?

Line 200 again ideally 'statistically' significant reductions for clarity throughout results

Discussion

Line 258 onwards – though some improvements were noted it would be important to acknowledge here at the outset that only short term changes were assessed and suggesting sustained change would require further justification. Further it is difficult to see how you can conclude that quality of care is preserved based on what you measured. You acknowledge later that euthanasia per se is a crude measure.

Line 277 onwards engagement with the portal – it would be valuable to acknowledge further that group access did not indicate increased individual vet access – this is mentioned in passing but could be clarified more. Further no posthoc comparisons are made so it is not clear that statistical significant improvement /difference was observed for both groups LG and HG.

Line 291-2 It would also be helpful to acknowledge that practice wide meetings were likely to engage more/most rather than all staff unless you have evidence to support this?

Line 299 – it remains difficult to see justification for suggesting 'sustained' reduction given your short followup.

Line 326 onwards = not central to your discussion but it would be useful to comment on the likely sustainability of the interventions you propose. Was there some feedback on the level of work and commitment from the group leads and practices?

Study limitations

The limitation could have been developed further and would merit a paragraph. As per the general comments above – further discussion of the fact that assessment was short term effect after some interventions – particularly the HG group could have been developed. Some of these would also be shorter than the 6 months proposed if their interventions are ongoing for much of that period. This could be clarified further and has implications for any discussion of sustainability. In the context of rolling these interventions out more broadly it would be helpful to comment on the sustainability and practicalities of undertaking these interventions (HG in particular) for an extended period of time - as well as the need to assess their effects for a longer period of time.

Strength and limitation to the approaches to communicating with practices and the challenges of reaching all vet could have been discussed also. Issues around multiple comparisons could also be acknowledged.

Figures and tables

Figure 2- the axes could also indicate these are percent of total rather than absolute frequencies.

It would be helpful to indicate the period of time in which the interventions occurred for HG and LG below the month axis potentially to clarify the duration of intervention activity relative to usage.

Table 1 – it would be informative to include results of any statistical comparisons in this table. This could be for key measures only – including vet and practice size measures

Tables 2 and 3 - It is not clear why the AM systemic and topical total %s appear to be greater than the total usage in each group – this would be helpful to just clarify – presumably this is multiple agents – topical+systemic- in a given consultation. This could be indicated in the legend.

Table 4 – some confidence intervals span one where their p values appear significant – this seems strange though may well reflect different analyses for IRR v the statistical comparisons – some may be worth checking – e.g. feb 2019 dogs.

Otherwise a nice paper

Best wishes

Dave Brodbelt, RVC

Reviewer #2 (Remarks to the Author):

The manuscript “A randomised controlled trial to reduce highest priority critically important antimicrobial prescription in companion animals” by Singleton et al. is a well-written manuscript and adds novel and valuable data to the important field of antimicrobial stewardship in veterinary medicine. The manuscript is technically sound and the presentation of the data and the length of the manuscript is adequate.

By using electronic prescription data from 60 companion animal practices, the authors evaluated the effect of two different educational intervention strategies on the frequency of prescription of highest priority critically important antimicrobials (HPCIA). The authors show that their intervention in the heavy intervention group (HIG) reduced the canine and feline HPCIA-prescribing consultancies significantly by 30% and 42%, respectively, but no effect was observed in the light intervention group when compared to the control group.

In my opinion, there is one major limitations of the study, which needs to be addressed by the authors. This concern is related to the definition of the intervention period in the HIG group: as indicated in the figure, the intervention started in March 2019, but in the HIG group, repeated reviews by the hub clinical lead took place until October 2019 (described in lines 167 ff). The follow-up data from the participating practices was collected until September 2019. The authors state that ‘Post-intervention monitoring was carried out between 1 April 2019 and 30 September 2019...’ (line 141), and no further follow-up data is provided. In my opinion, these monthly reviews by the hub clinical lead that took place until September 2019 are part of the intervention and should be defined as such (and also indicated in the figures). This however would mean that no follow-up data after this ‘intervention period’ in the HIG was presented in the study. The study would be clearly strengthened if the follow-up data would exceed this period. Otherwise the authors have to provide convincing evidence that the follow-up reviews were not perceived as intervention in the HIG. This would be a prerequisite to claim a sustained effect of the intervention in the HIG.

Other minor limitations: The set-up of the study does not allow to quantify the effect of the prescription change on patient outcome, because only the rate of euthanasia was assessed, which does not allow to draw any conclusions concerning patient outcome and safety. This limitation is however discussed by the authors in the manuscript. Another limitation lies in the fact that only practices from the CVS practice group were included, which limits the extrapolation of the results to other companion animal veterinary settings. This is also discussed as a limitation in the discussion part of the manuscript.

Detailed review:

Abstract:

Line 17: please specify what ‘0.5% of consultations’ refers to, to the reduction or the overall percentage of HPCIA-prescribing consultancies. Similar in line 18 for feline consultations, and in

line 19.

Line 19: "The LIG was not associated with significant variation". This sentence is unclear and needs rephrasing.

Introduction:

Line 25: The close proximity in which companion animals reside with humans is not a contributor to the development of AMR, so this sentence needs rephrasing. Factors that contribute to the development and selection of AMR in companion animal veterinary settings should be added here.

Line 32ff: Although cefovecin is an important HPCIA, especially in cats, fluoroquinolones should be mentioned here because they are very commonly prescribed in dogs, but also in cats.

Line 49ff: Two references here are probably not sufficient to cover the large number of studies that were performed on this topic. Similar, line 52f, add some studies that provided evidence.

Materials and Methods

Line 89: Not clear for the reader how these sites/branches were distributed between control and intervention groups and considered in the analyses.

Lines 117 and 123: Not clear whether these links belong to different videos, since the same SAVSNET site appears with both links.

Lines 125 ff: How did this review sessions take place and who participated on practice level. This is should be further specified.

Line 141: See comment above, the period of post-intervention monitoring should be reassessed.

Results

Line 180 and throughout the manuscript: please be consistent with the abbreviations (LG or LIG, HG or HIG)

Line 187ff: As mentioned above, please specify what is meant here with the percentage in parentheses. And could comparison to baseline data be added here for each group?

Lines 236f, 243: I think variation is not the correct term here.

Discussion:

Line 261: 'while preserving care quality' must be deleted, because this trial does not allow to evaluate the effect on care quality in the practices.

Line 271ff: This hypothesis/sentence should be switched to the introduction part of the manuscript.

Line 281 ff: I think the reason for this observation is that the intervention was still going in the four months follow-up period in the HIG group, see comments above.

Line 332: I think the follow-up period overall is relatively short and this limitation should be addressed in more detail in the discussion part of the manuscript.

Line 343: This finding is important and should be more critically discussed. A switch to potentiated aminopenicillins was suspected, and this class is classified as highly important antimicrobial by the WHO and known to be an important driver in the development of betalactam resistances f.ex. in Enterobacteriaceae.

Line 352: I think this statement is wrong, potentiated aminopenicillins (including amoxicillin) are among the most commonly prescribed antimicrobials in companion animal medicine.

Line 361: "non-infectious mediators for respiratory diseases" should be changed to "non-bacterial" or similar, because viral etiologies are assumed to be common.

Line 368ff: This sentence is very vague and needs specification.

Figures:

Figure 1: "8 practices not randomly selected": this is not mentioned in M&M and needs to be specified there. Furthermore, only 18 practices (2 declined to participate) completed post-intervention monitoring in the HIG group, so 20 needs to be changed to 18 here. A time scale on one site of the figure or a separate figure that indicates the time course of the intervention would be helpful.

Figure 2: Interesting that 4/18 HIG practices showed the highest prescription rate in dogs before intervention and this is not at all evident in the data (Table 2).

Figures 2b and 3b and others: As mentioned above, the time period of the intervention in the HIG should be reassessed.

Tables:

Table 1: could be switched to supplementary material.

Table 2: Typo 'upplementary material'. When referring to other tables, the content of these tables needs not to be specified in parentheses.

Table 4: The incident rate ratio is not further specified in any part of the Materials & Methods or Results section. If included, this should be specified. In the footnote indicated as "incidence rate ratio", please be consistent with the term.

References:

Several recent publications on antimicrobial stewardship in companion animal medicine should be added here, f.ex. Hopman et al 2018, 2019 and others.

REVIEWER COMMENTS

Reviewer #1 (Remarks to the Author):

General comments

This is a well written manuscript and valuable contribution to the literature on AM stewardship. Overall it is clear and presents the results well. Some general comments include –

- **Many thanks for providing such a comprehensive and helpful review of our manuscript. Thank you also for being willing to engage with us prior to resubmission of this work – this greatly increased our ability to provide a response and improve the manuscript, and for that we are very grateful. We have provided a response to each of your general and specific points below in bold. We have also highlighted any changes in the main text in yellow, and provided a line number where relevant and practical. As analyses has been repeated with a new extended dataset, some of our initial significance tests have changed (in either direction); to assist you in spotting these, we have highlighted any significant changes in red in the tables in the main manuscript.**

1. Interventions - these could have been described in more detail given these are central to the study. Some parts were but more details on the clinical lead review process and staff meeting processes would have been valuable.

- **Thank you for raising this important point. We have added further detail to this section of the methods as follows:**
 - **We have indicated that in the LIG that intervention material was addressed to the clinical director (CD) of each practice (lines 121-122), and that the CD could distribute these materials as they wished amongst staff (line 128-129). This approach was taken to continue the low intensity/efficiency ethos of the LIG.**
 - **We have also clarified that CDs were also initially approached in the HIG in lines 131-132 and 136.**
 - **We have also extended our explanation of the HIG intervention in lines 136-150. We hope this will have improved clarity, and would be happy to further clarify this intervention if you think it necessary.**

2. Post-intervention duration of effect– this cut-off indicates usage after the receipt of a letter for the two intervention groups with further implementations undertaken over subsequent months. This would appear to impact on the actual duration of interventions and period of effect after interventions have been completed. Some comment is made that the interventions were short term to 6 months. It would be beneficial to expand this in the discussion further as the interventions themselves extended over 1-3 months based on the results reported and though most practices would have had the major interventions near the onset of this 6 months, for the HG group – many may not have had the practice meetings and ongoing clinical reviews for some time after the intervention start date and as such the follow up ‘residual’ period could be quite a bit shorter in some practice. This might limit extrapolating too far beyond the actual intervention period of effect.

- **This is an excellent point, and thank you again for being willing to discuss this informally prior to resubmission. As agreed, we have now extended our post-initial intervention by two months to end of November 2019, and have repeated analyses to incorporate this extended time throughout. Although data are available to analyse a longer period of time**

post-intervention, we are cautious of entering into a period known to be affected by both a widescale acute vomiting outbreak in dogs (Dec 2019 – Mar 2020) and the impact of COVID-19 on veterinary practices (Mar 2020 onwards), either of which might perhaps unnecessarily complicate trial analyses and findings. Although this extension has enabled us to strengthen some of our findings, including a new finding that practices in the LIG now have a significant reduction in HPCIA prescription frequency in cats, we do agree that our initial submission suggested ‘sustained’ reduction too strongly; reference to this has been removed from the manuscript. We have reflected on the ‘new’ finding in lines 371-375, and have also provided a ‘warning’ against extrapolating too far in lines 375-380. As this change entailed extensive repeated analyses, we have simply highlighted changed findings in yellow in the manuscript, rather than pointing to every change by line. We have also changed all supplementary tables to reflect these changes.

2. Pre to post comparisons. The main statistical comparisons appear to be with the 3 treatment groups for the post-intervention AM usage as stated. It would have been interesting particularly for the main outcomes to compare to pre usage – not withstanding seasonal issues – was this an element of the statistical modelling? i.e. Changes over time in addition to between group change? This further element of the analysis could have been discussed in the discussion - or at least why it was not considered necessary.

- **Many thanks for this excellent suggestion. We have included an overall measure of temporal trend by each intervention group (rather than between intervention groups), with method explained in lines 182-184, and results in lines 243-248. As we have already reached the table/figure limit for this journal, we have included regression output values in supplementary material.**

Specific comments.

Abstract – page 2, line 12 clarify – presumably the EHR were used to identify high HPCIA prescribing practices which were then randomly allocated? (rather than the data used for allocation to 1 of the 3 groups)

- **You are correct, apologies for this lax phrasing which has been amended (lines 12-13).**

Line 14 – confirm control group received no direct interaction? This could be here also in the abstract.

- **You are correct, the CG received no direction interaction; this is noted in lines 154-156. Although we did initially note this in the abstract, word limit requirements forced its removal and we cannot easily see a way of re-including it without deleting other perhaps more essential parts of the abstract. We hope you are happy for us not to take this action?**

Line 18 – ideally include p values here. Clarify how also ‘maintenance’ of effect was assessed this may be harder to justify.

- **As above, we have removed reference to ‘maintenance’ of effect, and have instead included reference to feline LIG HPCIA reductions now being significant. We have also**

included p values, as requested, though have deleted specific reference to 'significant' to accommodate these within the word limit (lines 18 and 20).

Line 20 – significant reduction or variation – clarify?

- **Due to the above changes, this sentence has now been deleted.**

Conclusions- these could be more specific as to what interventions appear to be recommended.

- **Although we agree the abstract would benefit from more specific conclusions, word limit restrictions make this difficult without needing to remove other parts of the abstract. We hope this is OK?**

Introduction

Line 41 consider also Tompson PVM 2020 re qualitative approaches.

- **Many thanks for drawing my attention to this very interesting study, and for suggesting its inclusion; it has been cited (line 43).**

Method

Line 70. Consider clarifying that data related to animal and owner were de-identified, or similar.

- **This is a good point easily forgotten – apologies! We have included 'anonymised' on line 78.**

Line 93 – clarify if hpcia practice level median usage was reported, why only 66 practices were eligible? It would appear half of all 157 practices would be above the median practice level usage?

- **Apologies for trying to explain simply it now reads a little misleadingly. We divided practices into deciles, and selected practices if both dogs or cats were in the 40% highest decile and above (n=43). We then selected remaining practices which were in the 40% highest decile and above in dogs (n=8) or cats (n=17), that were also in the top 60% in the opposing species. We opted for this approach in order to identify practices that were 'generally' more frequent HPCIA prescribers, rather than biasing towards any particular species. We hope this makes sense, and we have clarified the text to explain this more fully (lines 99-105). Although not shown in this study, we did check that this method resulted in selection of practices which were more frequent HPCIA prescribers than those not selected for the trial, and this was found to be the case.**

Line 101 – clarify was this 'complete random allocation' simple random allocation, block random sampling etc?

- **This was simple random allocation – we have clarified this in line 112.**

Line 106 onwards. The details re the interventions could be expanded: Who received the

intervention practice email/letter and how was that planned to be communicated to vets within the practice.

- **Many thanks for this suggestion. As noted above, we have hopefully improved this section in lines 136-150.**

Line 125 – it would be helpful to provide more details of how clinical leads were selected within the CVS team, of what the initial reviews by the clinical lead consisted of as well as how and to who the clinical lead performed their followup ‘check ins’. Some more details could be provided in the main manuscript.

- **We have provided brief further details on hub clinical leads in lines 137-138, and additional detail on the review process in lines 136-150, thank you.**

Outcomes.

Line 142 – it would appear the primary outcome was percent of total consults with Hpcia. Reference to comparison to cg would appear unrequired here and could go in the statistical section.

- **Good point, we have removed this reference.**

Statistical analysis –

It would be useful to clarify that model assumptions and diagnostics/measures of fit were assessed and met (the latter in the results).

- **Many thanks for pointing out this omission. We have included reference to model assumptions and goodness of fit method in the methods (lines 184-185). We have included goodness of fit findings in supplementary tables, and have also provided reference to this in results (lines 230-232; 243-244; 254-256; 289).**

Were effects over time modelled relative to the time baseline t0 also? This was not clear. Could these have been considered within the modelling approach?

- **Many thanks for this suggestion – as noted above, we have now incorporated this idea into our analyses.**

Some incidence rates and ratios are reported – it would be good to clarify their calculation in the methods.

- **As discussed, on reflection we decided that these did not add a great deal to the paper, but did potentially introduce some confusion re. confidence intervals. We have now removed these from results tables.**

Were baseline characteristics compared statistically? This would help support similarity of groups at baseline.

- **Thank you for the suggestion. We have acted on this, detailing method of comparison in lines 174-175, a brief re-wording to increase strength of statement (as none were found to vary significantly) on line 207-208, and comparison findings in Table 1.**

Within the results some further linear regression methods appear to have been performed? These could be mentioned within the methods.

- **Many thanks for pointing this out – we haven't provided specific reference to the linear regression visualisation provided in figures 2 and 3, apologies. We have included a brief description of this in lines 185-187.**

Results

Participation, interventions and baseline data

Line 174 – though only 2 centres that did not take part, it would be interesting to know whether they were at an extreme of size / vet numbers – perhaps worth a comment here?

- **Thanks for this thought. Both of these practices varied in size, one tending towards the smaller end of the scale, and the other to the larger. However, neither could be considered an extreme. We do have feedback on why these practices declined to take part, but as this was given informally and in confidence we don't feel that we can add this to the manuscript. As such, in the interests of brevity we would like to keep this section as it stands – hope this is OK?**

Line 175 – consider comparing statistically the groups on key baseline measures – this would support further that they appeared similar.

- **Many thanks for this suggestion – we have acted on this as noted above.**

Line 178 – pre-intervention - it might be helpful to confirm these were not statistically different at baseline?

- **Many thanks for this great suggestion – we have acted on this in lines 214-215 and 217-218.**

More details of the response rates and participation in practice meetings also would have been helpful – though this would be in the results.

- **Thanks for this suggestion, we have acted on this as noted above.**

Line 180 – here you refer to LG, HG rather than LiG HiG – ideally make the same throughout.

- **Many thanks for pointing out this oversight; this has been corrected.**

Line 195 – it would be useful to confirm these practice decreases were statistically significant?

- **Thanks for this; we have addressed this suggestion above.**

Line 200 again ideally 'statistically' significant reductions for clarity throughout results

- **Many thanks, we have acted on this above.**

Discussion

Line 258 onwards – though some improvements were noted it would be important to acknowledge here at the outset that only short term changes were assessed and suggesting sustained change would require further justification. Further it is difficult to see how you can conclude that quality of care is preserved based on what you measured. You acknowledge later that euthanasia per se is a crude measure.

- **Thank you for these thoughts. We have addressed sustained change earlier in our response, and have removed reference to this in this section. We also agree that euthanasia alone is perhaps too crude to be able to fully justify stating preserved care quality. Hence, we have also deleted this claim.**

Line 277 onwards engagement with the portal – it would be valuable to acknowledge further that group access did not indicate increased individual vet access – this is mentioned in passing but could be clarified more. Further no posthoc comparisons are made so it is not clear that statistical significant improvement /difference was observed for both groups LG and HG.

- **Many thanks for pointing out this limitation, we have included a more explicit acknowledgement of this in lines 326-328. We have also decided to included a posthoc comparison following your comment, methods of which are in lines 192-193, and results in lines 296-298.**

Line 291-2 It would also be helpful to acknowledge that practice wide meetings were likely to engage more/most rather than all staff unless you have evidence to support this?

- **This is a good point thank you – we have amended accordingly (lines 325-326).**

Line 299 – it remains difficult to see justification for suggesting 'sustained' reduction given your short followup.

- **As above, we have now deleted any reference to sustained reductions throughout the manuscript.**

Line 326 onwards = not central to your discussion but it would be useful to comment on the likely sustainability of the interventions you propose. Was there some feedback on the level of work and commitment from the group leads and practices?

- **Many thanks for this suggestion. Although we did not formally request feedback (this would have been a good idea!), we have reflected in greater depth on sustainability in lines 371-375.**

Study limitations

The limitation could have been developed further and would merit a paragraph. As per the general comments above – further discussion of the fact that assessment was short term effect after some interventions – particularly the HG group could have been developed. Some of these would also be shorter than the 6 months proposed if their interventions are ongoing for much of that period. This could be clarified further and has implications for any discussion of sustainability. In the context of rolling these interventions out more broadly it would be helpful to comment on the sustainability and practicalities of undertaking these interventions (HG in particular) for an extended period of time - as well as the need to assess their effects for a longer period of time.

Strength and limitation to the approaches to communicating with practices and the challenges of reaching all vet could have been discussed also. Issues around multiple comparisons could also be acknowledged.

- **Many thanks for these suggestions. We believe we have now covered these limitations throughout the text in this revision. However, if you feel we have missed any or not covered these in enough depth we would be more than happy to amend further.**

Figures and tables

Figure 2- the axes could also indicate these are percent of total rather than absolute frequencies. It would be helpful to indicate the period of time in which the interventions occurred for HG and LG below the month axis potentially to clarify the duration of intervention activity relative to usage.

- **Many thanks for these useful suggestions; we have acted on both, also extending these to figure 3.**

Table 1 – it would be informative to include results of any statistical comparisons in this table. This could be for key measures only – including vet and practice size measures

- **Many thanks for this suggestion, which we have acted upon. For completeness we have made comparisons on all measures included in this table.**

Tables 2 and 3 - It is not clear why the AM systemic and topical total %s appear to be greater than the total usage in each group – this would be helpful to just clarify – presumably this is multiple agents – topical+systemic- in a given consultation. This could be indicated in the legend.

- **This is a good point – you are correct in your assumption, thanks for pointing it out. We have included a brief clarification in tables 2 and 3.**

Table 4 – some confidence intervals span one where their p values appear significant – this seems strange though may well reflect different analyses for IRR v the statistical comparisons – some may be worth checking – e.g. feb 2019 dogs.

- **As discussed above, we have decided to remove IRRs from these tables.**

Otherwise a nice paper

Best wishes

Dave Brodbelt, RVC

Thank you again for the time you have taken to provide this excellent review. We really appreciate the effort you have gone too, and feel that your contribution has greatly increased the quality of this work.

**Best wishes,
David**

Reviewer #2 (Remarks to the Author):

The manuscript “A randomised controlled trial to reduce highest priority critically important antimicrobial prescription in companion animals” by Singleton et al. is a well-written manuscript and adds novel and valuable data to the important field of antimicrobial stewardship in veterinary medicine. The manuscript is technically sound and the presentation of the data and the length of the manuscript is adequate.

By using electronic prescription data from 60 companion animal practices, the authors evaluated the effect of two different educational intervention strategies on the frequency of prescription of highest priority critically important antimicrobials (HPCIAAs). The authors show that their intervention in the heavy intervention group (HIG) reduced the canine and feline HPCIAA-prescribing consultancies significantly by 30% and 42%, respectively, but no effect was observed in the light intervention group when compared to the control group.

- **Many thanks for providing such a comprehensive and helpful review of our manuscript – this has greatly increased our ability to provide a response and improve the manuscript, and for that we are very grateful. We have provided a response to each of your general and specific points below in bold. We have also highlighted any changes in the main text in yellow, and provided a line number where relevant and practical. As analyses has been repeated with a new extended dataset, some of our initial significance tests have changed (in either direction); to assist you in spotting these, we have highlighted any significant changes in red in the tables in the main manuscript.**

In my opinion, there is one major limitations of the study, which needs to be addressed by the authors. This concern is related to the definition of the intervention period in the HIG group: as indicated in the figure, the intervention started in March 2019, but in the HIG group, repeated reviews by the hub clinical lead took place until October 2019 (described in lines 167 ff). The follow-up data from the participating practices was collected until September 2019. The authors state that ‘Post-intervention monitoring was carried out between 1 April 2019 and 30 September 2019...’ (line 141), and no further follow-up data is provided. In my opinion, these monthly reviews by the hub clinical lead that took place until September 2019 are part of the intervention and should be defined as such (and also indicated in the figures). This however would mean that no follow-up data after this ‘intervention period’ in the HIG was presented in the study. The study would be clearly strengthened if the follow-up data would exceed this period. Otherwise the authors have to provide

convincing evidence that the follow-up reviews were not perceived as intervention in the HIG. This would be a prerequisite to claim a sustained effect of the intervention in the HIG.

- **This is an excellent point, thank you. We agree that the post-benchmarking intervention support programme could be perceived as a continued intervention, and we like your suggestion of extending follow-up to at least partially ameliorate this limitation. As such, we have now extended our post-initial intervention by two months to end of November 2019, and have repeated analyses to incorporate this extended time throughout. Although data are available to analyse a longer period of time post-intervention, we are cautious of entering into a period known to be affected by a widescale acute vomiting outbreak in dogs in the UK (Dec 2019 – Mar 2020) and the wide ranging impact of COVID-19 on veterinary practices (Mar 2020 onwards), either of which might perhaps unnecessarily complicate trial analyses and findings. Although this extension has enabled us to strengthen some of our findings, including a new finding that practices in the LIG have now be associated with a significant reduction in HPCIA prescription frequency in cats, we do still agree that our initial submission suggested ‘sustained’ reduction too strongly; reference to this has been removed from the manuscript. We have reflected on the ‘new’ finding in lines 371-375, and have also provided a ‘warning’ against extrapolating too far in 375-380. As this change entailed extensive repeated analyses, we have simply highlighted changed findings in yellow in the manuscript, rather than pointing to every change by line. We have also changed all supplementary tables to reflect these changes.**

Other minor limitations: The set-up of the study does not allow to quantify the effect of the prescription change on patient outcome, because only the rate of euthanasia was assessed, which does not allow to draw any conclusions concerning patient outcome and safety. This limitation is however discussed by the authors in the manuscript. Another limitation lies in the fact that only practices from the CVS practice group were included, which limits the extrapolation of the results to other companion animal veterinary settings. This is also discussed as a limitation in the discussion part of the manuscript.

- **Many thanks for these observations, we agree that these are limitations and we are glad you feel we have adequately addressed this in the discussion. The other reviewer additionally felt that a claim of ours relating to retaining ‘care quality’ at the beginning of the discussion was too strong considering the relative crudeness of euthanasia frequency. We have agreed to remove this in the revised manuscript.**

Detailed review:

Abstract:

Line 17: please specify what ‘0.5% of consultations’ refers to, to the reduction or the overall percentage of HPCIA-prescribing consultancies. Similar in line 18 for feline consultations, and in line 19.

- **Many thanks for pointing out this lax use of language. We currently are pushed against the abstract limit so have opted for now to just add the word ‘total’ to hopefully clarify this point – we hope you feel this is adequate (line 17).**

Line 19: “The LIG was not associated with significant variation’. This sentence is unclear and needs rephrasing.

- **Thanks for your thoughts on this. Due to the changes mentioned above, the LIG now has been associated with a specific reduction in cats, so we have amended this sentence accordingly (lines 19-20).**

Introduction:

Line 25: The close proximity in which companion animals reside with humans is not a contributor to the development of AMR, so this sentence needs rephrasing. Factors that contribute to the development and selection of AMR in companion animal veterinary settings should be added here.

- **Great point, this was poor phrasing on our part. We have added an acknowledgement of antimicrobial prescription being the primary driver of AMR at this point, which hopefully clarifies that this refers to development, whereas close proximity relates to transmission (lines 25-26). To reduce repetition, we have also removed reference to prescription being the primary driver of AMR later on in the paragraph, and have moved the related references to this earlier point in the paragraph.**

Line 32ff: Although cefovecin is an important HPCIA, especially in cats, fluoroquinolones should be mentioned here because they are very commonly prescribed in dogs, but also in cats.

- **Thank you for this suggestion. We have added an acknowledgement of fluoroquinolones in lines 36-37. In this case we have opted for ‘relatively frequently prescribed’ as although we agree that as with cefovecin this group are likely too frequently prescribed, frequency of prescription of this class is somewhat lower than cefovecin, which should be considered the ‘main’ HPCIA problem.**

Line 49ff: Two references here are probably not sufficient to cover the large number of studies that were performed on this topic. Similar, line 52f, add some studies that provided evidence.

- **We have added additional references here to more fully encompass past work, thanks (lines 53-54).**

Materials and Methods

Line 89: Not clear for the reader how these sites/branches were distributed between control and intervention groups and considered in the analyses.

- **Thank you for this thought. We have added further clarity into how practices were selected for inclusion in the trial in lines 99-105. Site distribution is shown in lines 111-112.**

Lines 117 and 123: Not clear whether these links belong to different videos, since the same SAVSNET site appears with both links.

- **While each link points to a site of the same appearance, and two of the same videos appear in each, the HIG webpage contains an additional video which explains how the additional in-depth benchmarking report might be interpreted. We have added a clarification of this in line 134.**

Lines 125 ff: How did this review sessions take place and who participated on practice level. This is should be further specified.

- **Many thanks for this suggestion. We have included further explanation of this process in lines 136-150.**

Line 141: See comment above, the period of post-intervention monitoring should be reassessed.

- **As discussed above, we have now lengthened the post-intervention monitoring period, thank you.**

Results

Line 180 and throughout the manuscript: please be consistent with the abbreviations (LG or LIG, HG or HIG)

- **Apologies for this oversight. We have corrected these and reviewed the rest of the manuscript to ensure we haven't made mistakes elsewhere.**

Line 187ff: As mentioned above, please specify what is meant here with the percentage in parentheses. And could comparison to baseline data be added here for each group?

- **Many thanks for this suggestion – we have added explanations in parentheses in lines 223, 224, 225, 227, 228 and 229. Regarding comparison to baseline, we do cover this in lines 211-220. We did initially consider a more combined approach to presenting results, but found that this raised potential for confusion as opposed to the more segmented approach we have opted for here. However, we are happy to revisit this if you feel it would greatly benefit presentation of results.**

Lines 236f, 243: I think variation is not the correct term here.

- **This is a good point, thank you. We have changed this to 'reduction' in both cases (lines 281 and 287).**

Discussion:

Line 261: 'while preserving care quality' must be deleted, because this trial does not allow to evaluate the effect on care quality in the practices.

- **As noted above, we agree this claim was too strong and it has been deleted accordingly.**

Line 271ff: This hypothesis/sentence should be switched to the introduction part of the manuscript.

- **This is an excellent point, thank you – we had not clearly signposted this hypothesis in the introduction, and have added this in lines 57-63. As we feel this point has now been covered in the manuscript we have deleted the paragraph in discussion, but added to a sentence to ensure that the social norm point does have some representation in the discussion (lines 311-313).**

Line 281 ff: I think the reason for this observation is that the intervention was still going in the four months follow-up period in the HIG group, see comments above.

- **Thank you, as above we have amended the analyses and manuscript to acknowledge this.**

Line 332: I think the follow-up period overall is relatively short and this limitation should be addressed in more detail in the discussion part of the manuscript.

- **Thank you, as above we have amended the analyses and manuscript to acknowledge this, and have discussed at greater length in the discussion (lines 375-380).**

Line 343: This finding is important and should be more critically discussed. A switch to potentiated aminopenicillins was suspected, and this class is classified as highly important antimicrobial by the WHO and known to be an important driver in the development of betalactam resistances f.ex. in Enterobacteriaceae.

- **Thank you for this – we were unsure to what degree to discuss this finding, as we felt lack of significance does arguably limit how far we might be able to push this point. We have included your suggestion to include reference to ‘high’ importance by the WHO (lines 399-400), and have added further discussion in lines 400-403.**

Line 352: I think this statement is wrong, potentiated aminopenicillins (including amoxicillin) are among the most commonly prescribed antimicrobials in companion animal medicine.

- **This statement relates to human medical practice, not companion animal medicine, and so is correct (at least in the UK). We recognise though that this might be misread by others, so have changed phrasing here, also briefly acknowledging that infrequent human use has arisen due to pressure to reduce dependence on amoxiclav (line 400).**

Line 361: “non-infectious mediators for respiratory diseases” should be changed to “non-bacterial” or similar, because viral etiologies are assumed to be common.

- **Absolutely, thanks for pointing this out – we have adjusted accordingly (line 410).**

Line 368ff: This sentence is very vague and needs specification.

- **Apologies we aren’t completely sure which sentence you are referring to here, but we have added an example into the sentence we think you are meaning (lines 417-418).**

Figures:

Figure 1: “8 practices not randomly selected”: this is not mentioned in M&M and needs to be specified there. Furthermore, only 18 practices (2 declined to participate) completed post-intervention monitoring in the HIG group, so 20 needs to be changed to 18 here. A time scale on one site of the figure or a separate figure that indicates the time course of the intervention would be helpful.

- **Thank you, we do refer to these 8 non-selected practices in methods (lines 110). As data continued to be available for all practices regardless of their involvement in the trial, post-intervention monitoring was completed for all 20 practices in the HIG group, with analyses being on an intention-to-treat basis. We have provided a clarification of this in lines 200-202. Thank you for your suggestion of a timescale on a figure – we have included this in figures 2 and 3.**

Figure 2: Interesting that 4/18 HIG practices showed the highest prescription rate in dogs before intervention and this is not at all evident in the data (Table 2).

- **Yes, this is interesting. It is possible that although these practices might display a higher prescription rate, they might have contributed fewer consultations, thereby being less represented in findings. We also calculated proportions + CIs to take account of clustering, thereby to some degree ameliorating the potential impact of relatively extreme values on population-level summaries.**

Figures 2b and 3b and others: As mentioned above, the time period of the intervention in the HIG should be reassessed.

- **Thank you, we have addressed this comment above.**

Tables:

Table 1: could be switched to supplementary material.

- **Thank you. It is a requirement of CONSORT guidelines (which is a prerequisite for publication in this journal) to display baseline characteristics in the manuscript. As such, this table must stay in the main document.**

Table 2: Typo ‘supplementary material’. When referring to other tables, the content of these tables needs not to be specified in parentheses.

- **Many thanks for noticing this, and apologies for this oversight – it has been corrected.**

Table 4: The incident rate ratio is not further specified in any part of the Materials & Methods or Results section. If included, this should be specified. In the footnote indicated as “incidence rate ratio”, please be consistent with the term.

- **Thank you. Following some discussion with the other reviewer who felt their inclusion could introduce confusion for readers, we have decided to remove incidence rate ratios from this manuscript.**

References:

Several recent publications on antimicrobial stewardship in companion animal medicine should be added here, f.ex. Hopman et al 2018, 2019 and others.

- **Thank you for this suggestion, particularly the Hopman works which we were not fully aware of previously. We have added this and other previous publications to this manuscript.**

Thank you again for the time you have taken to provide this excellent review. We really appreciate the effort you have gone to, and feel that your contribution has greatly increased the quality of this work.

**Best wishes,
David**

REVIEWERS' COMMENTS

Reviewer #1 (Remarks to the Author):

The authors have revised the manuscript and clarified it greatly. All main queries have been addressed - some minor details could be checked as below.

Abstract

These changes are fine – it would be useful to add 'HPCIA' after feline – to clarify that it was HPCIA rather than all AM usage.

Results -

Line 231 – it would be helpful for consistency also to include 'HPCIA usage' for clarity – after 'For cats, a significant 39.0% and 16.7% decrease...'

line 246 also 'The LIG was associated with a significant reduction in cats in..' clarify this is reduction of HPCIA usage in cats – rather than reduction in cats per se...

line 289 - presumably here you were looking for significant increase in culture and susceptibility testing rather than decrease after intervention?

supplementary materials for simplicity personally I would not refer to actual table numbers in supplementary materials but just suppl materials themselves – I will leave this with the editors.

Reviewer #2 (Remarks to the Author):

Thank you for providing a revised manuscript. I believe this is a greatly improved manuscript and relevant to support antimicrobial stewardship activities in veterinary medicine. All points raised in the review have been addressed. There are two minor remarks concerning the abstract in the revised manuscript:

I would suggest to make some changes in the wording (f.ex. shorten introduction) so that the follow-up period is still included.

The term 'feline reduction' (line 20) should be changed to 'reduction in cats' or similar.

Otherwise I have no further remarks.

REVIEWER COMMENTS

Reviewer #1 (Remarks to the Author):

The authors have revised the manuscript and clarified it greatly. All main queries have been addressed - some minor details could be checked as below.

Many thanks for your continued help in improving the quality of this manuscript – it is much appreciated. As before, we have included our response in bold below your thoughts, and provided a line number where relevant.

Abstract

These changes are fine – it would be useful to add 'HPCIA' after feline – to clarify that it was HPCIA rather than all AM usage.

Many thanks for this suggestion. The editorial team have allowed us to extend the abstract up to 200 words if needed, so we have returned some of the earlier excluded detail to enhance readability, including your suggestion here (Lines 23-24).

Results -

Line 231 – it would be helpful for consistency also to include 'HPCIA usage' for clarity – after 'For cats, a significant 39.0% and 16.7% decrease...'

Thank you for this suggestion - we have included this (line 233), and to enhance clarity have also added in the same phrase in lines 239 and 241.

line 246 also 'The LIG was associated with a significant reduction in cats in..' clarify this is reduction of HPCIA usage in cats – rather than reduction in cats per se...

Thank you, we have added this in (line 249), and have also added clarifications in line 253-254, 260 and 262.

line 289 - presumably here you were looking for significant increase in culture and susceptibility testing rather than decrease after intervention?

Good point, thank you for noticing this omission. We have corrected this to 'increase' (line 288), and have also corrected the following paragraph (line 295).

supplementary materials for simplicity personally I would not refer to actual table numbers in supplementary materials but just suppl materials themselves – I will leave this with the editors.

Thank you, the editor has requested that table numbers are retained, so these have been left in the manuscript.

Thank you again for your really useful advice with regards to this manuscript!

Reviewer #2 (Remarks to the Author):

Thank you for providing a revised manuscript. I believe this is a greatly improved manuscript and relevant to support antimicrobial stewardship activities in veterinary medicine. All points raised in the review have been addressed. There are two minor remarks concerning the abstract in the revised manuscript:

Many thanks for your continued help in improving the quality of this manuscript – it is much appreciated. As before, we have included our response in bold below your thoughts, and provided a line number where relevant.

I would suggest to make some changes in the wording (f.ex. shorten introduction) so that the follow-up period is still included.

Many thanks for this suggestion. The editorial team have allowed us to extend the abstract up to 200 words if needed, so we have returned some of the earlier excluded detail to enhance standalone readability, including your suggestion here (Lines 18-20).

The term 'feline reduction' (line 20) should be changed to 'reduction in cats' or similar.

Many thanks for this suggestion – we have changed this accordingly, and added a small clarification of the denominator in the following brackets (lines 23-24).

Otherwise I have no further remarks.

Thank you again for your really useful advice with regards to this manuscript!